# INSTAGAN:
# INSTANCE-AWARE IMAGE-TO-IMAGE TRANSLATION

**Sangwoo Mo**[*], **Minsu Cho**[†], **Jinwoo Shin**[*,‡]
[*]Korea Advanced Institute of Science and Technology (KAIST), Daejeon, Korea
[†]Pohang University of Science and Technology (POSTECH), Pohang, Korea
[‡]AItrics, Seoul, Korea
[*]{swmo, jinwoos}@kaist.ac.kr, [†]mscho@postech.ac.kr

## ABSTRACT

Unsupervised image-to-image translation has gained considerable attention due to
the recent impressive progress based on generative adversarial networks (GANs).
However, previous methods often fail in challenging cases, in particular, when
an image has *multiple* target instances and a translation task involves significant
changes in *shape*, *e.g.,* translating pants to skirts in fashion images. To tackle
the issues, we propose a novel method, coined *instance-aware GAN (InstaGAN)*,
that incorporates the *instance* information (*e.g.,* object segmentation masks) and
improves *multi-instance transfiguration*. The proposed method translates both an
image and the corresponding set of instance attributes while maintaining the per-
mutation invariance property of the instances. To this end, we introduce a context
preserving loss that encourages the network to learn the identity function outside
of target instances. We also propose a sequential mini-batch inference/training
technique that handles multiple instances with a limited GPU memory and en-
hances the network to generalize better for multiple instances. Our comparative
evaluation demonstrates the effectiveness of the proposed method on different im-
age datasets, in particular, in the aforementioned challenging cases. Code and
results are available in https://github.com/sangwoomo/instagan.

## 1 INTRODUCTION

Cross-domain generation arises in many machine learning tasks, including neural machine trans-
lation (Artetxe et al., 2017; Lample et al., 2017), image synthesis (Reed et al., 2016; Zhu et al.,
2016), text style transfer (Shen et al., 2017), and video generation (Bansal et al., 2018; Wang et al.,
2018a; Chan et al., 2018). In particular, the unpaired (or unsupervised) image-to-image translation
has achieved an impressive progress based on variants of generative adversarial networks (GANs)
(Zhu et al., 2017; Liu et al., 2017; Choi et al., 2017; Almahairi et al., 2018; Huang et al., 2018;
Lee et al., 2018), and has also drawn considerable attention due to its practical applications includ-
ing colorization (Zhang et al., 2016), super-resolution (Ledig et al., 2017), semantic manipulation
(Wang et al., 2018b), and domain adaptation (Bousmalis et al., 2017; Shrivastava et al., 2017; Hoff-
man et al., 2017). Previous methods on this line of research, however, often fail on challenging
tasks, in particular, when the translation task involves significant changes in *shape* of instances (Zhu
et al., 2017) or the images to translate contains *multiple* target instances (Gokaslan et al., 2018). Our
goal is to extend image-to-image translation towards such challenging tasks, which can strengthen
its applicability up to the next level, *e.g.,* changing pants to skirts in fashion images for a customer
to decide which one is better to buy. To this end, we propose a novel method that incorporates the
*instance* information of multiple target objectsin the framework of generative adversarial networks
(GAN); hence we called it *instance-aware GAN (InstaGAN)*. In this work, we use the object seg-
mentation masks for instance information, which may be a good representation for instance shapes,
as it contains object boundaries while ignoring other details such as color. Using the information,
our method shows impressive results for *multi-instance transfiguration* tasks, as shown in Figure 1.

Our main contribution is three-fold: an instance-augmented neural architecture, a context preserving
loss, and a sequential mini-batch inference/training technique. First, we propose a neural network
architecture that translates both an image and the corresponding set of instance attributes. Our ar-
chitecture can translate an arbitrary number of instance attributes conditioned by the input, and is
designed to be permutation-invariant to the order of instances. Second, we propose a context preserv-

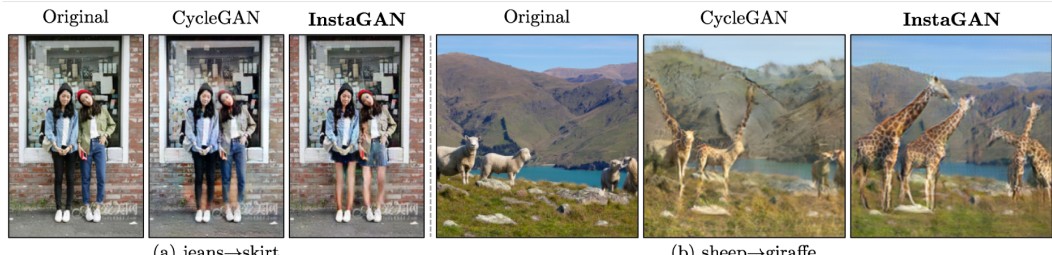

Figure 1: Translation results of the prior work (CycleGAN, Zhu et al. (2017)), and our proposed method, InstaGAN. Our method shows better results for multi-instance transfiguration problems.

ing loss that encourages the network to focus on target instances in translation and learn an identity function outside of them. Namely, it aims at preserving the background context while transforming the target instances. Finally, we propose a sequential mini-batch inference/training technique, *i.e.,* translating the mini-batches of instance attributes sequentially, instead of doing the entire set at once. It allows to handle a large number of instance attributes with a limited GPU memory, and thus enhances the network to generalize better for images with many instances. Furthermore, it improves the translation quality of images with even a few instances because it acts as data augmentation during training by producing multiple intermediate samples. All the aforementioned contributions are dedicated to how to incorporates the instance information (*e.g.,* segmentation masks) for image-to-image translation. However, we believe that our approach is applicable to numerous other cross-domain generation tasks where set-structured side information is available.

To the best of our knowledge, we are the first to report image-to-image translation results for multi-instance transfiguration tasks. A few number of recent methods (Kim et al., 2017; Liu et al., 2017; Gokaslan et al., 2018) show some transfiguration results but only for images with a single instance often in a clear background. Unlike the previous results in a simple setting, our focus is on the *harmony* of instances naturally rendered with the background. On the other hand, CycleGAN (Zhu et al., 2017) show some results for multi-instance cases, but report only a limited performance for transfiguration tasks. At a high level, the significance of our work is also on discovering that the instance information is effective for shape-transforming image-to-image translation, which we think would be influential to other related research in the future. Mask contrast-GAN (Liang et al., 2017) and Attention-GAN (Mejjati et al., 2018) use segmentation masks or predicted attentions, but only to attach the background to the (translated) cropped instances. They do not allow to transform the shapes of the instances. To the contrary, our method learns how to preserve the background by optimizing the context preserving loss, thus facilitating the shape transformation.

## 2  INSTAGAN: INSTANCE-AWARE IMAGE-TO-IMAGE TRANSLATION

Given two image domains $\mathcal{X}$ and $\mathcal{Y}$, the problem of image-to-image translation aims to learn mappings across different image domains, $G_{XY} : \mathcal{X} \rightarrow \mathcal{Y}$ or/and $G_{YX} : \mathcal{Y} \rightarrow \mathcal{X}$, *i.e.,* transforming target scene elements while preserving the original contexts. This can also be formulated as a conditional generative modeling task where we estimate the conditionals $p(y|x)$ or/and $p(x|y)$. The goal of *unsupervised* translation we tackle is to recover such mappings only using unpaired samples from marginal distributions of original data, $p_{\texttt{data}}(x)$ and $p_{\texttt{data}}(y)$ of two image domains.

The main and unique idea of our approach is to incorporate the additional *instance* information, *i.e.,* augment a space of *set of instance attributes* $\mathcal{A}$ to the original image space $\mathcal{X}$, to improve the image-to-image translation. The set of instance attributes $\boldsymbol{a} \in \mathcal{A}$ comprises all individual attributes of $N$ target instances: $\boldsymbol{a} = \{a_i\}_{i=1}^{N}$. In this work, we use an instance segmentation mask only, but we remark that any useful type of instance information can be incorporated for the attributes. Our approach then can be described as learning joint-mappings between attribute-augmented spaces $\mathcal{X} \times \mathcal{A}$ and $\mathcal{Y} \times \mathcal{B}$. This leads to disentangle different instances in the image and allows the generator to perform an accurate and detailed translation. We learn our attribute-augmented mapping in the framework of generative adversarial networks (GANs) (Goodfellow et al., 2014), hence, we call it *instance-aware GAN (InstaGAN)*. We present details of our approach in the following subsections.

### 2.1  INSTAGAN ARCHITECTURE

Recent GAN-based methods (Zhu et al., 2017; Liu et al., 2017) have achieved impressive performance in the unsupervised translation by jointly training two coupled mappings $G_{XY}$ and $G_{YX}$ with

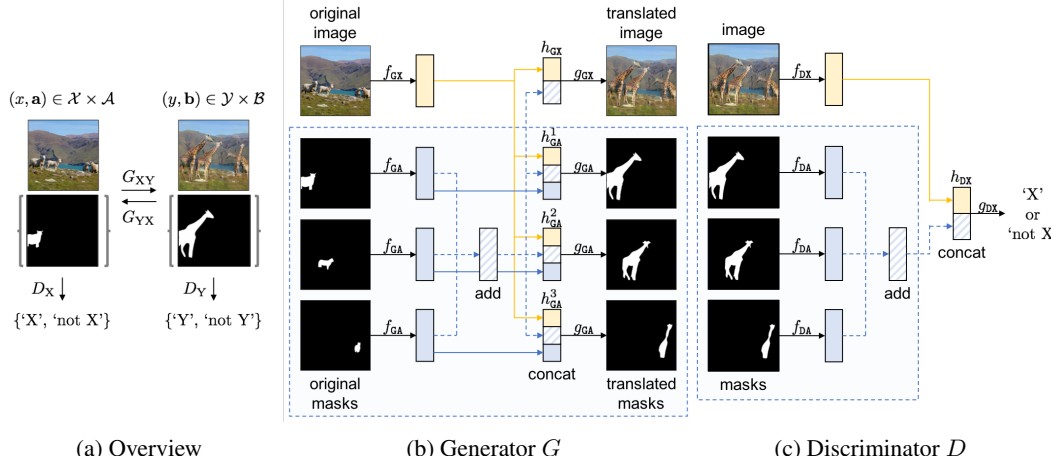

(a) Overview          (b) Generator $G$          (c) Discriminator $D$

Figure 2: (a) Overview of InstaGAN, where generators $G_{XY}$, $G_{YX}$ and discriminator $D_X$, $D_Y$ follows the architectures in (b) and (c), respectively. Each network is designed to encode both an image and set of instance masks. $G$ is permutation equivariant, and $D$ is permutation invariant to the set order. To achieve properties, we sum features of all set elements for invariance, and then concatenate it with the identity mapping for equivariance.

a cycle-consistency loss that encourages $G_{YX}(G_{XY}(x)) \approx x$ and $G_{XY}(G_{YX}(y)) \approx y$. Namely, we choose to leverage the CycleGAN approach (Zhu et al., 2017) to build our InstaGAN. However, we remark that training two coupled mappings is not essential for our method, and one can also design a single mapping following other approaches (Benaim & Wolf, 2017; Galanti et al., 2018). Figure 2 illustrates the overall architecture of our model. We train two coupled generators $G_{XY} : \mathcal{X} \times \mathcal{A} \rightarrow \mathcal{Y} \times \mathcal{B}$ and $G_{YX} : \mathcal{Y} \times \mathcal{B} \rightarrow \mathcal{X} \times \mathcal{A}$, where $G_{XY}$ translates the original data $(x, \boldsymbol{a})$ to the target domain data $(y', \boldsymbol{b}')$ (and vice versa for $G_{YX}$), with adversarial discriminators $D_X : \mathcal{X} \times \mathcal{A} \rightarrow \{\text{'X'}, \text{'not X'}\}$ and $D_Y : \mathcal{Y} \times \mathcal{B} \rightarrow \{\text{'Y'}, \text{'not Y'}\}$, where $D_X$ determines if the data (original $(x, \boldsymbol{a})$ or translated $(x', \boldsymbol{a}')$) is in the target domain $\mathcal{X} \times \mathcal{A}$ or not (and vice versa for $D_Y$).

Our generator $G$ encodes both $x$ and $\boldsymbol{a}$, and translates them into $y'$ and $\boldsymbol{b}'$. Notably, the order of the instance attributes in the set $\boldsymbol{a}$ should not affect the translated image $y'$, and each instance attribute in the set $\boldsymbol{a}$ should be translated to the corresponding one in $\boldsymbol{b}'$. In other words, $y'$ is *permutation-invariant* with respect to the instances in $\boldsymbol{a}$, and $\boldsymbol{b}'$ is *permutation-equivariant* with respect to them. These properties can be implemented by introducing proper operators in feature encoding (Zaheer et al., 2017). We first extract *individual* features from image and attributes using image feature extractor $f_{GX}$ and attribute feature extractor $f_{GA}$, respectively. The attribute features individually extracted using $f_{GA}$ are then aggregated into a permutation-invariant set feature via summation: $\sum_{i=1}^{N} f_{GA}(a_i)$. As illustrated in Figure 2b, we concatenate some of image and attribute features with the set feature, and feed them to image and attribute generators. Formally, the image representation $h_{GX}$ and the $n$-th attribute representation $h_{GA}^n$ in generator $G$ can be formulated as:

$$h_{GX}(x, \boldsymbol{a}) = \left[ f_{GX}(x); \sum_{i=1}^{N} f_{GA}(a_i) \right], \quad h_{GA}^n(x, \boldsymbol{a}) = \left[ f_{GX}(x); \sum_{i=1}^{N} f_{GA}(a_i); f_{GA}(a_n) \right], \quad (1)$$

where each attribute encoding $h_{GA}^n$ process features of all attributes as a contextual feature. Finally, $h_{GX}$ is fed to the image generator $g_{GX}$, and $h_{GA}^n$ ($n = 1, \ldots, N$) are to the attribute generator $g_{GA}$.

On the other hand, our discriminator $D$ encodes both $x$ and $\boldsymbol{a}$ (or $x'$ and $\boldsymbol{a}'$), and determines whether the pair is from the domain or not. Here, the order of the instance attributes in the set $\boldsymbol{a}$ should not affect the output. In a similar manner above, our representation in discriminator $D$, which is permutation-invariant to the instances, is formulated as:

$$h_{DX}(x, \boldsymbol{a}) = \left[ f_{DX}(x); \sum_{i=1}^{N} f_{DA}(a_i) \right], \quad (2)$$

which is fed to an adversarial discriminator $g_{DX}$.

We emphasize that the joint encoding of both image $x$ and instance attributes $\boldsymbol{a}$ for each neural component is crucial because it allows the network to learn the *relation* between $x$ and $\boldsymbol{a}$. For

example, if two separate encodings and discriminators are used for $x$ and $\boldsymbol{a}$, the generator may be misled to produce image and instance masks that do not match with each other. By using the joint encoding and discriminator, our generator can produce an image of instances properly depicted on the area consistent with its segmentation masks. As will be seen in Section 3, our approach can disentangle output instances considering their original layouts. Note that any types of neural networks may be used for sub-network architectures mentioned above such as $f_{\text{GX}}$, $f_{\text{GA}}$, $f_{\text{DX}}$, $f_{\text{DA}}$, $g_{\text{GX}}$, $g_{\text{GA}}$, and $g_{\text{DX}}$. We describe the detailed architectures used in our experiments in Appendix A.

## 2.2 TRAINING LOSS

Remind that an image-to-image translation model aims to translate a domain while keeping the original contexts (*e.g.,* background or instances' domain-independent characteristics such as the looking direction). To this end, we both consider the *domain* loss, which makes the generated outputs to follow the style of a target domain, and the *content* loss, which makes the outputs to keep the original contents. Following our baseline model, CycleGAN (Zhu et al., 2017), we use the GAN loss for the domain loss, and consider both the cycle-consistency loss (Kim et al., 2017; Yi et al., 2017) and the identity mapping loss (Taigman et al., 2016) for the content losses.[1] In addition, we also propose a new content loss, coined *context preserving loss*, using the original and predicted segmentation information. In what follows, we formally define our training loss in detail. For simplicity, we denote our loss function as a function of a single training sample $(x, \boldsymbol{a}) \in \mathcal{X} \times \mathcal{A}$ and $(y, \boldsymbol{b}) \in \mathcal{Y} \times \mathcal{B}$, while one has to minimize its empirical means in training.

The GAN loss is originally proposed by Goodfellow et al. (2014) for generative modeling via alternately training generator $G$ and discriminator $D$. Here, $D$ determines if the data is a real one of a fake/generated/translated one made by $G$. There are numerous variants of the GAN loss (Nowozin et al., 2016; Arjovsky et al., 2017; Li et al., 2017; Mroueh et al., 2017), and we follow the LSGAN scheme (Mao et al., 2017), which is empirically known to show a stably good performance:

$$\mathcal{L}_{\text{LSGAN}} = (D_{\text{X}}(x, \boldsymbol{a}) - 1)^2 + D_{\text{X}}(G_{\text{YX}}(y, \boldsymbol{b}))^2 + (D_{\text{Y}}(y, \boldsymbol{b}) - 1)^2 + D_{\text{Y}}(G_{\text{XY}}(x, \boldsymbol{a}))^2. \quad (3)$$

For keeping the original content, the cycle-consistency loss $\mathcal{L}_{\text{cyc}}$ and the identity mapping loss $\mathcal{L}_{\text{idt}}$ enforce samples not to lose the original information after translating twice and once, respectively:

$$\mathcal{L}_{\text{cyc}} = \|G_{\text{YX}}(G_{\text{XY}}(x, \boldsymbol{a})) - (x, \boldsymbol{a})\|_1 + \|G_{\text{XY}}(G_{\text{YX}}(y, \boldsymbol{b})) - (y, \boldsymbol{b})\|_1, \quad (4)$$

$$\mathcal{L}_{\text{idt}} = \|G_{\text{XY}}(y, \boldsymbol{b}) - (y, \boldsymbol{b})\|_1 + \|G_{\text{YX}}(x, \boldsymbol{a}) - (x, \boldsymbol{a})\|_1. \quad (5)$$

Finally, our newly proposed context preserving loss $\mathcal{L}_{\text{ctx}}$ enforces to translate instances only, while keeping outside of them, *i.e.,* background. Formally, it is a pixel-wise weighted $\ell_1$-loss where the weight is 1 for background and 0 for instances. Here, note that backgrounds for two domains become different in transfiguration-type translation involving significant shape changes. Hence, we consider the non-zero weight only if a pixel is in background in both original and translated ones. Namely, for the original samples $(x, \boldsymbol{a})$, $(y, \boldsymbol{b})$ and the translated one $(y', \boldsymbol{b}')$, $(x', \boldsymbol{a}')$, we let the weight $w(\boldsymbol{a}, \boldsymbol{b}')$, $w(\boldsymbol{b}, \boldsymbol{a}')$ be one minus the element-wise minimum of binary represented instance masks, and we propose

$$\mathcal{L}_{\text{ctx}} = \|w(\boldsymbol{a}, \boldsymbol{b}') \odot (x - y')\|_1] + \|w(\boldsymbol{b}, \boldsymbol{a}') \odot (y - x')\|_1 \quad (6)$$

where $\odot$ is the element-wise product. In our experiments, we found that the context preserving loss not only keeps the background better, but also improves the quality of generated instance segmentations. Finally, the total loss of InstaGAN is

$$\mathcal{L}_{\text{InstaGAN}} = \underbrace{\mathcal{L}_{\text{LSGAN}}}_{\text{GAN (domain) loss}} + \underbrace{\lambda_{\text{cyc}}\mathcal{L}_{\text{cyc}} + \lambda_{\text{idt}}\mathcal{L}_{\text{idt}} + \lambda_{\text{ctx}}\mathcal{L}_{\text{ctx}}}_{\text{content loss}}, \quad (7)$$

where $\lambda_{\text{cyc}}, \lambda_{\text{idt}}, \lambda_{\text{ctx}} > 0$ are some hyper-parameters balancing the losses.

## 2.3 SEQUENTIAL MINI-BATCH TRANSLATION

While the proposed architecture is able to translate an arbitrary number of instances in principle, the GPU memory required linearly increases with the number of instances. For example, in our experiments, a machine was able to forward only a small number (say, 2) of instance attributes

---

[1] We remark that the identity mapping loss is also used in CycleGAN (see Figure 9 of Zhu et al. (2017)).

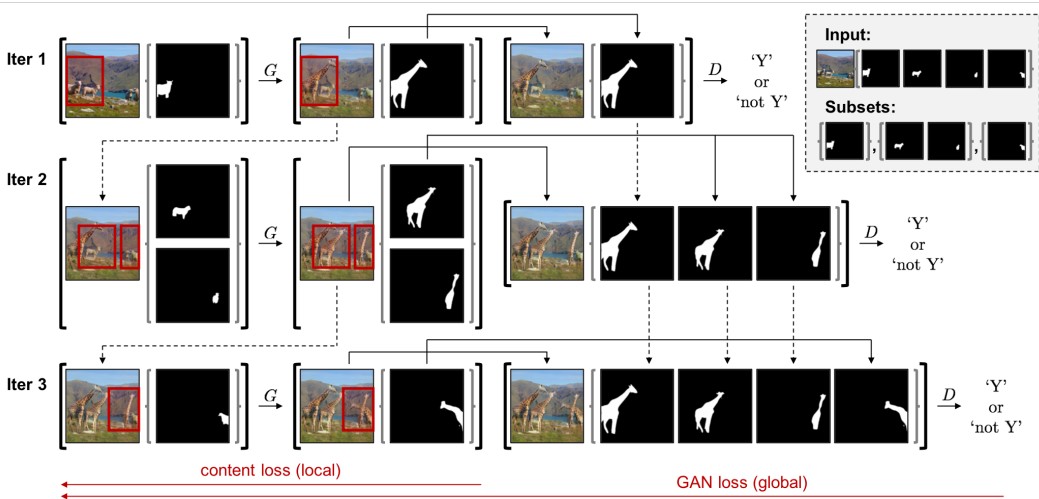

Figure 3: Overview of the sequential mini-batch training with instance subsets (mini-batches) of size 1,2, and 1, as shown in the top right side. The content loss is applied to the intermediate samples of current mini-batch, and GAN loss is applied to the samples of aggregated mini-batches. We detach every iteration in training, in that the real line indicates the backpropagated paths and dashed lines indicates the detached paths. See text for details.

during training, and thus the learned model suffered from poor generalization to images with a larger number of instances. To address this issue, we propose a new inference/training technique, which allows to train an arbitrary number of instances without increasing the GPU memory. We first describe the sequential inference scheme that translates the subset of instances sequentially, and then describe the corresponding mini-batch training technique.

Given an input $(x, \boldsymbol{a})$, we first divide the set of instance masks $\boldsymbol{a}$ into mini-batches $\boldsymbol{a}_1, \ldots, \boldsymbol{a}_M$, *i.e.*, $\boldsymbol{a} = \bigcup_i \boldsymbol{a}_i$ and $\boldsymbol{a}_i \cap \boldsymbol{a}_j = \emptyset$ for $i \neq j$. Then, at the $m$-th iteration for $m = 1, 2, \ldots, M$, we translate the image-mask pair $(x_m, \boldsymbol{a}_m)$, where $x_m$ is the translated image $y'_{m-1}$ from the previous iteration, and $x_1 = x$. In this sequential scheme, at each iteration, the generator $G$ outputs an intermediate translated image $y'_m$, which accumulates all mini-batch translations up to the current iteration, and a translated mini-batch of instance masks $\boldsymbol{b}'_m$:

$$(y'_m, \boldsymbol{b}'_m) = G(x_m, \boldsymbol{a}_m) = G(y'_{m-1}, \boldsymbol{a}_m). \tag{8}$$

In order to align the translated image with mini-batches of instance masks, we aggregate all the translated mini-batch and produce a translated sample:

$$(y'_m, \boldsymbol{b}'_{1:m}) = (y'_m, \cup_{i=1}^m \boldsymbol{b}'_i). \tag{9}$$

The final output of the proposed sequential inference scheme is $(y'_M, \boldsymbol{b}'_{1:M})$.

We also propose the corresponding sequential training algorithm, as illustrated in Figure 3. We apply content loss (4-6) to the intermediate samples $(y'_m, \boldsymbol{b}'_m)$ of current mini-batch $\boldsymbol{a}_m$, as it is just a function of inputs and outputs of the generator $G$.[2] In contrast, we apply GAN loss (3) to the samples of aggregated mini-batches $(y'_m, \boldsymbol{b}'_{1:m})$, because the network fails to align images and masks when using only a partial subset of instance masks. We used real/original samples $\{x\}$ with the full set of instance masks only. Formally, the sequential version of the training loss of InstaGAN is

$$\mathcal{L}_{\texttt{InstaGAN-SM}} = \sum_{m=1}^{M} \mathcal{L}_{\texttt{LSGAN}}((x, \boldsymbol{a}), (y'_m, \boldsymbol{b}'_{1:m})) + \mathcal{L}_{\texttt{content}}((x_m, \boldsymbol{a}_m), (y'_m, \boldsymbol{b}'_m)) \tag{10}$$

where $\mathcal{L}_{\texttt{content}} = \lambda_{\texttt{cyc}} \mathcal{L}_{\texttt{cyc}} + \lambda_{\texttt{idt}} \mathcal{L}_{\texttt{idt}} + \lambda_{\texttt{ctx}} \mathcal{L}_{\texttt{ctx}}$.

We detach every $m$-th iteration of training, *i.e.*, backpropagating with the mini-batch $\boldsymbol{a}_m$, so that only a fixed GPU memory is required, regardless of the number of training instances.[3] Hence, the

---

[2] The cycle-consistency loss (4) needs reconstruction sample $(x''_m, \boldsymbol{a}''_m)$. However, it is just a twice translated current mini-batch sample, *i.e.*, for the opposite direction generator $G'$, $(x''_m, \boldsymbol{a}''_m) = G'(G(x_m, \boldsymbol{a}_m))$.

[3] We still recommend users to increase the subset size as long as the GPU memory allows. This is because too many sequential steps may hurt the permutation-invariance property of our model.

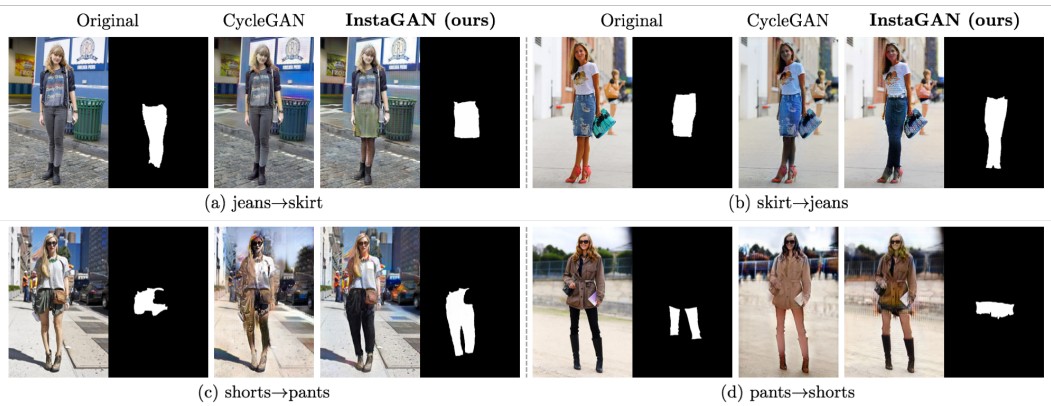

Figure 4: Translation results on clothing co-parsing (CCP) (Yang et al., 2014) dataset.

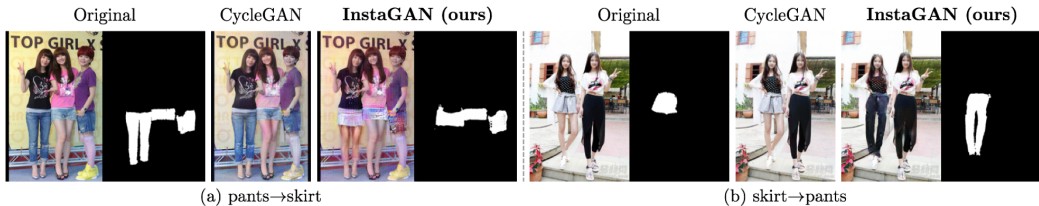

Figure 5: Translation results on multi-human parsing (MHP) (Zhao et al., 2018) dataset.

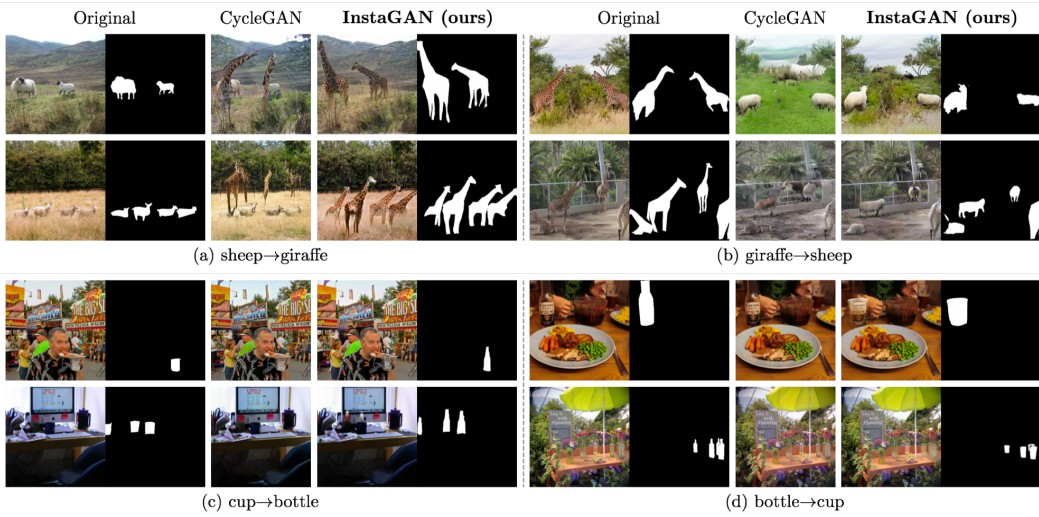

Figure 6: Translation results on COCO (Lin et al., 2014) dataset.

sequential training allows for training with samples containing many instances, and thus improves the generalization performance. Furthermore, it also improves translation of an image even with a few instances, compared to the one-step approach, due to its data augmentation effect using intermediate samples $(x_m, \boldsymbol{a}_m)$. In our experiments, we divided the instances into mini-batches $\boldsymbol{a}_1, \ldots, \boldsymbol{a}_M$ according to the decreasing order of the spatial sizes of instances. Interestingly, the decreasing order showed a better performance than the random order. We believe that this is because small instances tend to be occluded by other instances in images, thus often losing their intrinsic shape information.

## 3 EXPERIMENTAL RESULTS

### 3.1 IMAGE-TO-IMAGE TRANSLATION RESULTS

We first qualitatively evaluate our method on various datasets. We compare our model, InstaGAN, with the baseline model, CycleGAN (Zhu et al., 2017). For fair comparisons, we doubled the number of parameters of CycleGAN, as InstaGAN uses two networks for image and masks, respectively. We sample two classes from various datasets, including clothing co-parsing (CCP) (Yang et al.,

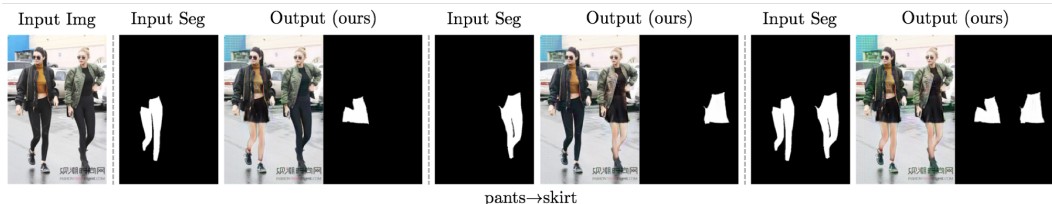

| Input Img | Input Seg | Output (ours) | Input Seg | Output (ours) | Input Seg | Output (ours) |

pants→skirt

Figure 7: Results of InstaGAN varying over different input masks.

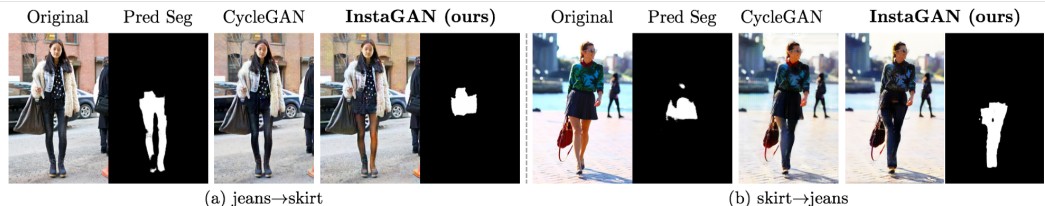

| Original | Pred Seg | CycleGAN | **InstaGAN (ours)** | Original | Pred Seg | CycleGAN | **InstaGAN (ours)** |

(a) jeans→skirt · (b) skirt→jeans

Figure 8: Translation results on CCP dataset, using predicted mask for inference.

2014), multi-human parsing (MHP) (Zhao et al., 2018), and MS COCO (Lin et al., 2014) datasets, and use them as the two domains for translation. In visualizations, we merge all instance masks into one for the sake of compactness. See Appendix B for detailed settings for our experiments. The translation results for three datasets are presented in Figure 4, 5, and 6, respectively. While CycleGAN mostly fails, our method generates reasonable shapes of the target instances and keeps the original contexts by focusing on the instances via the context preserving loss. For example, see the results on sheep↔giraffe in Figure 6. CycleGAN often generates sheep-like instances but loses the original background. InstaGAN not only generates better sheep or giraffes, but also preserves the layout of the original instances, *i.e.,* the looking direction (left, right, front) of sheep and giraffes are consistent after translation. More experimental results are presented in Appendix E. Code and results are available in https://github.com/sangwoomo/instagan.

On the other hand, our method can control the instances to translate by conditioning the input, as shown in Figure 7. Such a control is impossible under CycleGAN. We also note that we focus on complex (multi-instance transfiguration) tasks to emphasize the advantages of our method. Nevertheless, our method is also attractive to use even for simple tasks (*e.g.,* horse↔zebra) as it reduces false positives/negatives via the context preserving loss and enables to control translation. We finally emphasize that our method showed good results even when we use predicted segmentation for inference, as shown in Figure 8, and this can reduce the cost of collecting mask labels in practice.[4]

Finally, we also quantitatively evaluate the translation performance of our method. We measure the classification score, the ratio of images predicted as the target class by a pretrained classifier. Specifically, we fine-tune the final layers of the ImageNet (Deng et al., 2009) pretrained VGG-16 (Simonyan & Zisserman, 2014) network, as a binary classifier for each domain. Table 1 and Table 2 in Appendix D show the classification scores for CCP and COCO datasets, respectively. Our method outperforms CycleGAN in all classification experiments, *e.g.,* ours achieves 23.2% accuracy for the pants→shorts task, while CycleGAN obtains only 8.5%.

## 3.2 ABLATION STUDY

We now investigate the effects of each component of our proposed method in Figure 9. Our method is composed of the InstaGAN architecture, the context preserving loss $\mathcal{L}_{\text{ctx}}$, and the sequential mini-batch inference/training technique. We progressively add each component to the baseline model, CycleGAN (with doubled parameters). First, we study the effect of our architecture. For fair comparison, we train a CycleGAN model with an additional input channel, which translates the mask-augmented image, hence we call it CycleGAN+Seg. Unlike our architecture which translates the set of instance masks, CycleGAN+Seg translates the union of all masks at once. Due to this, CycleGAN+Seg fails to translate some instances and often merge them. On the other hand, our architecture keeps every instance and disentangles better. Second, we study the effect of the context

---

[4] For the results in Figure 8, we trained a pix2pix (Isola et al., 2017) model to predict a single mask from an image, but one can also utilize recent methods to predict instance masks in supervised (He et al., 2017) or weakly-supervised (Zhou et al., 2018) way.

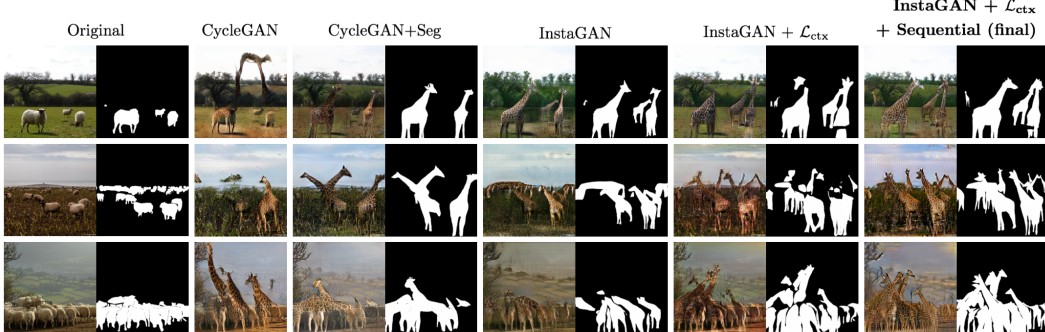

Figure 9: Ablation study on the effect of each component of our method: the InstaGAN architecture, the context preserving loss, and the sequential mini-batch inference/training algorithm, which are denoted as InstaGAN, $\mathcal{L}_{\text{ctx}}$, and Sequential, respectively.

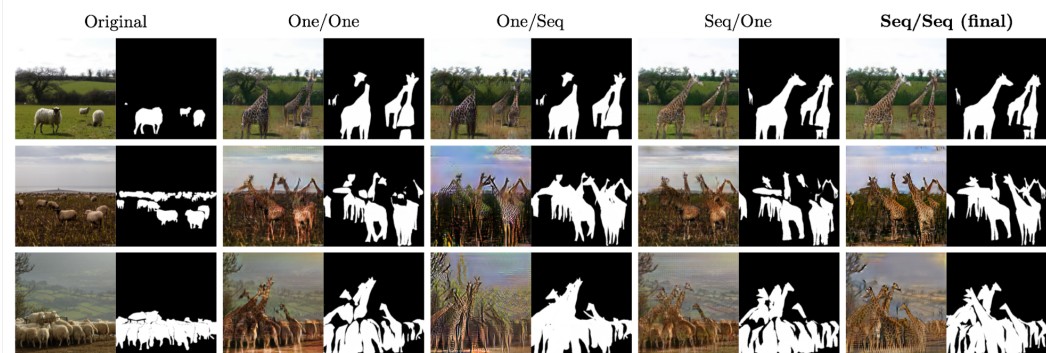

Figure 10: Ablation study on the effects of the sequential mini-batch inference/training technique. The left and right side of title indicates which method used for training and inference, respectively, where "One" and "Seq" indicate the one-step and sequential schemes, respectively.

preserving loss: it not only preserves the background better (row 2), but also improves the translation results as it regularizes the mapping (row 3). Third, we study the effect of our sequential translation: it not only improves the generalization performance (row 2,3) but also improves the translation results on few instances, via data augmentation (row 1).

Finally, Figure 10 reports how much the sequential translation, denoted by "Seq", is effective in inference and training, compared to the one-step approach, denoted by "One". For the one-step training, we consider only two instances, as it is the maximum number affordable for our machines. On the other hand, for the sequential training, we sequentially train two instances twice, *i.e.,* images of four instances. For the one-step inference, we translate the entire set at once, and for the sequential inference, we sequentially translate two instances at each iteration. We find that our sequential algorithm is effective for both training and inference: (a) training/inference = One/Seq shows blurry results as intermediate data have not shown during training and stacks noise as the iteration goes, and (b) Seq/One shows poor generalization performance for multiple instances as the one-step inference for many instances is not shown in training (due to a limited GPU memory).

## 4 CONCLUSION

We have proposed a novel method incorporating the set of instance attributes for image-to-image translation. The experiments on different datasets have shown successful image-to-image translation on the challenging tasks of multi-instance transfiguration, including new tasks, *e.g.,* translating jeans to skirt in fashion images. We remark that our ideas utilizing the set-structured side information have potential to be applied to other cross-domain generations tasks, *e.g.,* neural machine translation or video generation. Investigating new tasks and new information could be an interesting research direction in the future.

ACKNOWLEDGMENTS

This work was supported by the National Research Council of Science & Technology (NST) grant by the Korea government (MSIP) (No. CRC-15-05-ETRI), by the ICT R&D program of MSIT/IITP [2016-0-00563, Research on Adaptive Machine Learning Technology Development for Intelligent Autonomous Digital Companion], and also by Basic Science Research Program (NRF-2017R1E1A1A01077999) through the National Research Foundation of Korea (NRF) funded by the Ministry of Science, ICT.

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

## A    ARCHITECTURE DETAILS

We adopted the network architectures of CycleGAN (Zhu et al., 2017) as the building blocks for our proposed model. In specific, we adopted ResNet 9-blocks generator (Johnson et al., 2016; He et al., 2016) and PatchGAN (Isola et al., 2017) discriminator. ResNet generator is composed of downsampling blocks, residual blocks, and upsampling blocks. We used downsampling blocks and residual blocks for encoders, and used upsampling blocks for generators. On the other hand, PatchGAN discriminator is composed of 5 convolutional layers, including normalization and non-linearity layers. We used the first 3 convolution layers for feature extractors, and the last 2 convolution layers for classifier. We preprocessed instance segmentation as a binary foreground/background mask, hence simply used it as an 1-channel binary image. Also, since we concatenated two or three features to generate the final outputs, we doubled or tripled the input dimension of those architectures. Similar to prior works (Johnson et al., 2016; Zhu et al., 2017), we applied Instance Normalization (IN) (Ulyanov & Lempitsky, 2016) for both generators and discriminators. In addition, we observed that applying Spectral Normalization (SN) (Miyato et al., 2018) for discriminators significantly improves the performance, although we used LSGAN (Mao et al., 2017), while the original motivation of SN was to enforce Lipschitz condition to match with the theory of WGAN (Arjovsky et al., 2017; Gulrajani et al., 2017). We also applied SN for generators as suggested in Self-Attention GAN (Zhang et al., 2018), but did not observed gain for our setting.

## B    TRAINING DETAILS

For all the experiments, we simply set $\lambda_{cyc} = 10$, $\lambda_{idt} = 10$, and $\lambda_{ctx} = 10$ for our loss (7). We used Adam (Kingma & Ba, 2014) optimizer with batch size 4, training with 4 GPUs in parallel. All networks were trained from scratch, with learning rate of 0.0002 for $G$ and 0.0001 for $D$, and $\beta_1 = 0.5$, $\beta_2 = 0.999$ for the optimizer. Similar to CycleGAN (Zhu et al., 2017), we kept learning rate for first 100 epochs and linearly decayed to zero for next 100 epochs for multi-human parsing (MHP) (Zhao et al., 2018) and COCO (Lin et al., 2014) dataset, and kept learning rate for first 400 epochs and linearly decayed for next 200 epochs for clothing co-parsing (CCP) (Yang et al., 2014) dataset, as it contains smaller number of samples. We sampled two classes from the datasets above, and used it as two domains for translation. We resized images with size 300×200 (height×width) for CCP dataset, 240×160 for MHP dataset, and 200×200 for COCO dataset, respectively.

## C    TREND OF TRANSLATION RESULTS

We tracked the trend of translation results over epoch increases, as shown in Figure 11. Both image and mask smoothly adopted to the target instances. For example, the remaining parts in legs slowly disappears, and the skirt slowly constructs the triangular shapes.

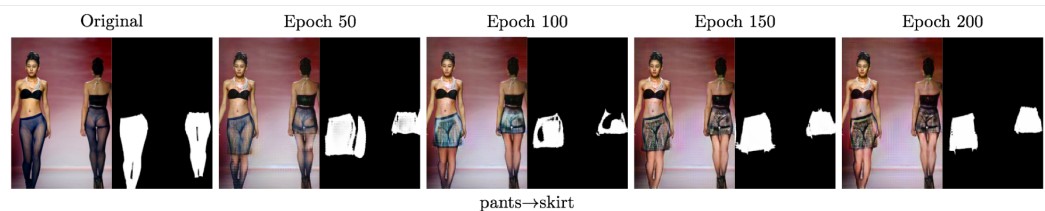

Figure 11: Trend of the translation results of our method over epoch increases.

# D    QUANTITATIVE RESULTS

We evaluated the classification score for CCP and COCO dataset. Unlike CCP dataset, COCO dataset suffers from the false positive problem, that the classifier fails to determine if the generator produced target instances on the right place. To overcome this issue, we measured the masked classification score, where the input images are masked by the corresponding segmentations. We note that CycleGAN and our method showed comparable results for the naïve classification score, but ours outperformed for the masked classification score, as it reduces the false positive problem.

Table 1: Classification score for CCP dataset.

|  | jeans→skirt | | skirt→jeans | | shorts→pants | | pants→shorts | |
|---|---|---|---|---|---|---|---|---|
|  | train | test | train | test | train | test | train | test |
| Real | 0.970 | 0.888 | 0.982 | 0.946 | 1.000 | 0.984 | 0.990 | 0.720 |
| CycleGAN | 0.465 | 0.371 | 0.561 | 0.483 | 0.845 | 0.524 | 0.305 | 0.085 |
| **InstaGAN (ours)** | **0.665** | **0.600** | **0.658** | **0.540** | **0.898** | **0.768** | **0.373** | **0.232** |

Table 2: Classification score (masked) for COCO dataset.

|  | sheep→giraffe | | giraffe→sheep | | cup→bottle | | bottle→cup | |
|---|---|---|---|---|---|---|---|---|
|  | train | test | train | test | train | test | train | test |
| Real | 0.891 | 0.911 | 0.925 | 0.930 | 0.746 | 0.723 | 0.622 | 0.566 |
| CycleGAN | 0.313 | 0.594 | 0.291 | 0.512 | 0.368 | 0.403 | 0.290 | 0.275 |
| **InstaGAN (ours)** | **0.406** | **0.781** | **0.355** | **0.642** | **0.443** | **0.465** | **0.322** | **0.333** |

# E    MORE TRANSLATION RESULTS

We present more qualitative results in high resolution images.

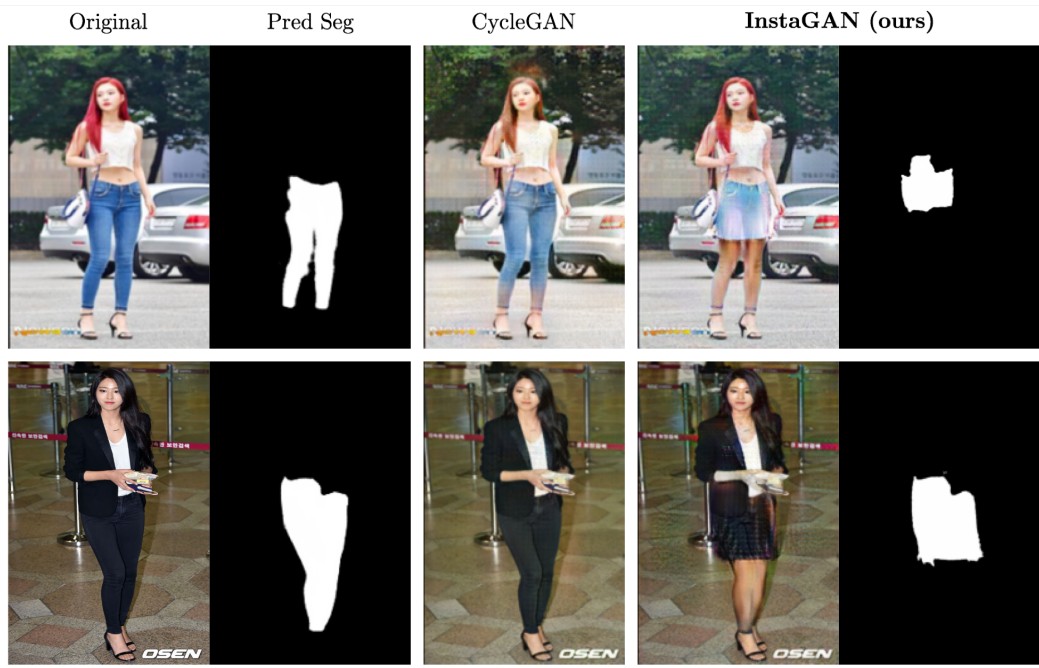

Figure 12: Translation results for images searched from Google to test the generalization performance of our model. We used a pix2pix (Isola et al., 2017) model to predict the segmentation.

| Original | CycleGAN | **InstaGAN (ours)** |
|---|---|---|

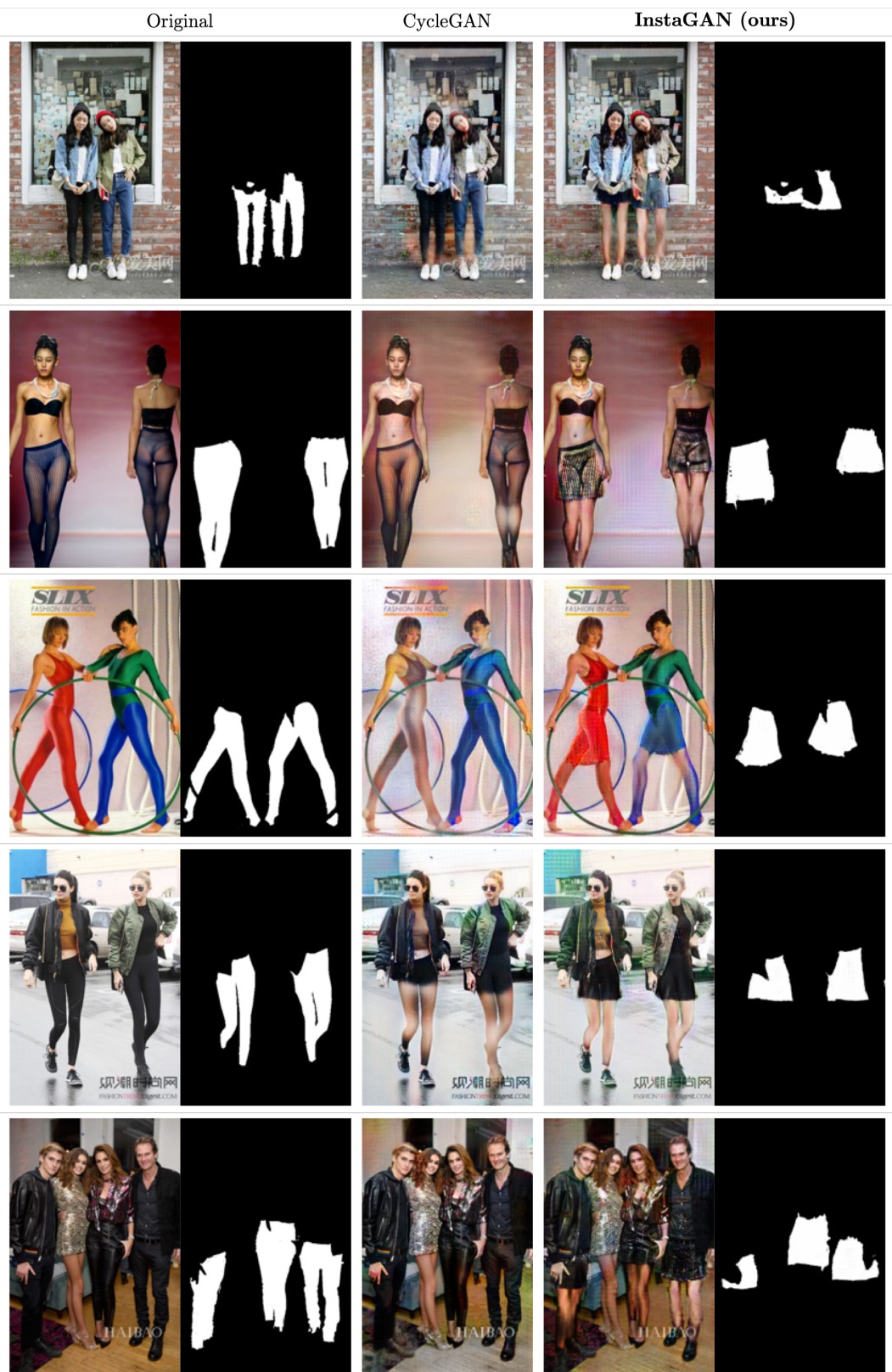

Figure 13: More translation results on MHP dataset (pants→skirt).

| Original | CycleGAN | **InstaGAN (ours)** |
|---|---|---|

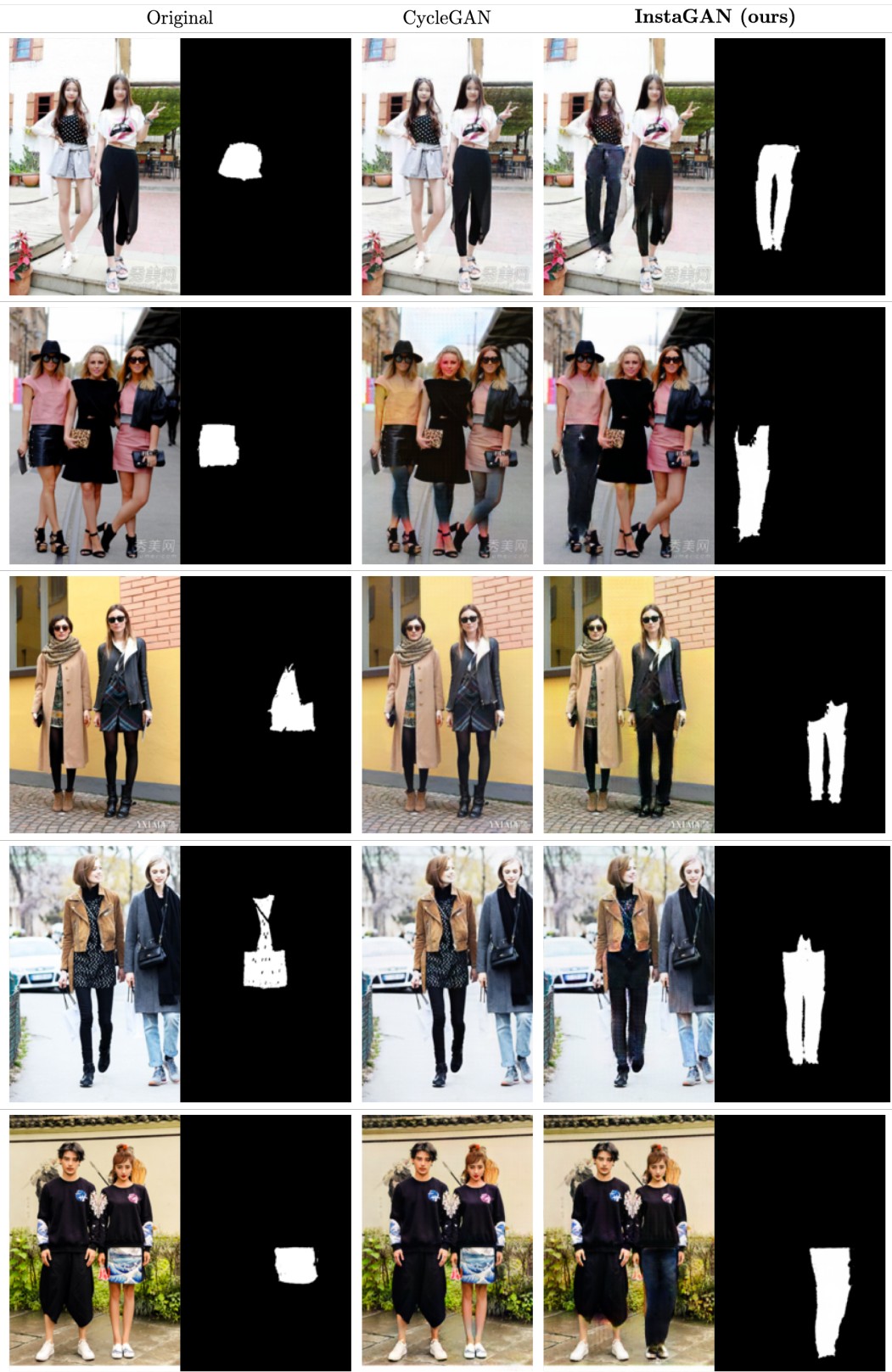

Figure 14: More translation results on MHP dataset (skirt→pants).

Original                     CycleGAN                     **InstaGAN (ours)**

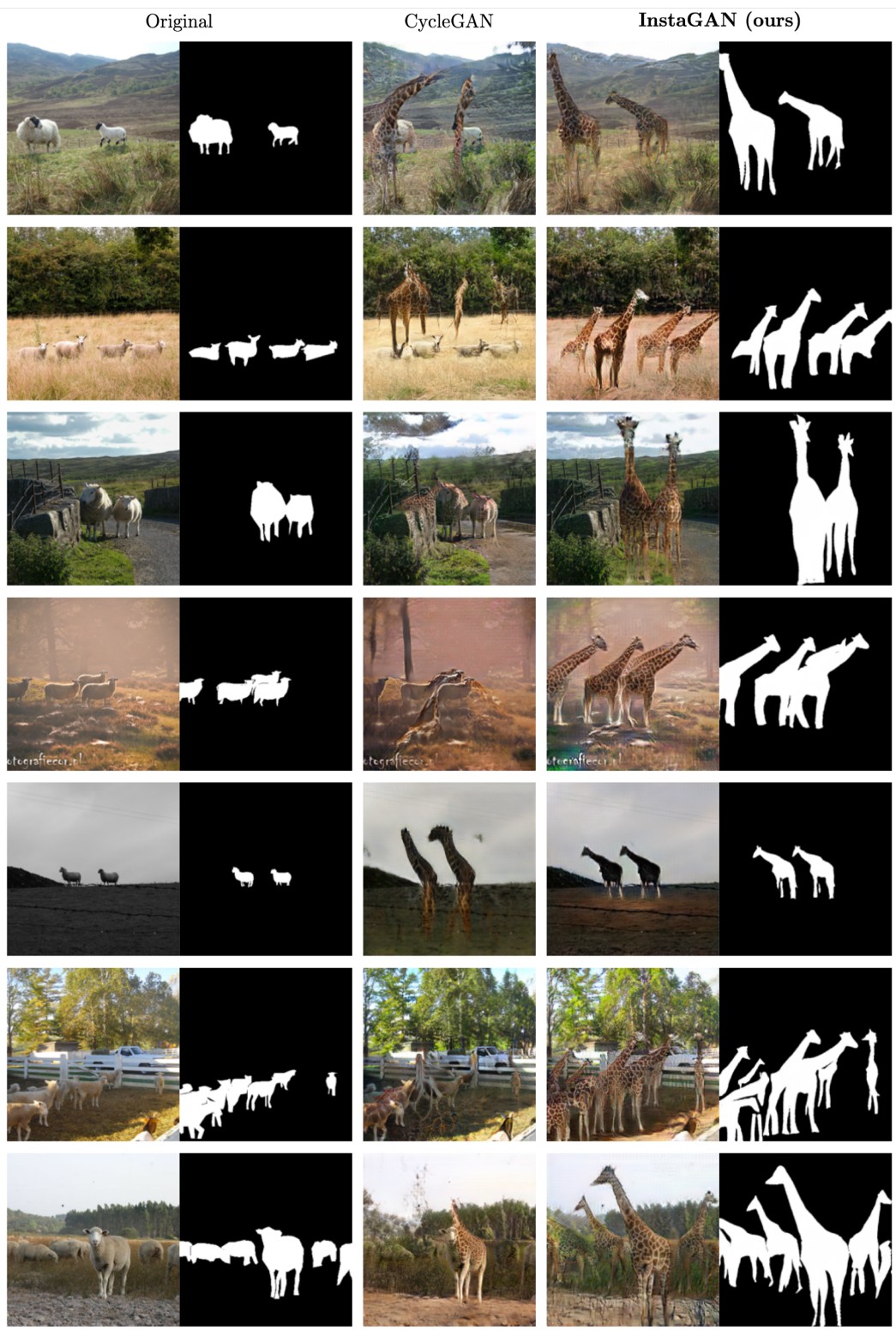

Figure 15: More translation results on COCO dataset (sheep→giraffe).

Original CycleGAN **InstaGAN (ours)**

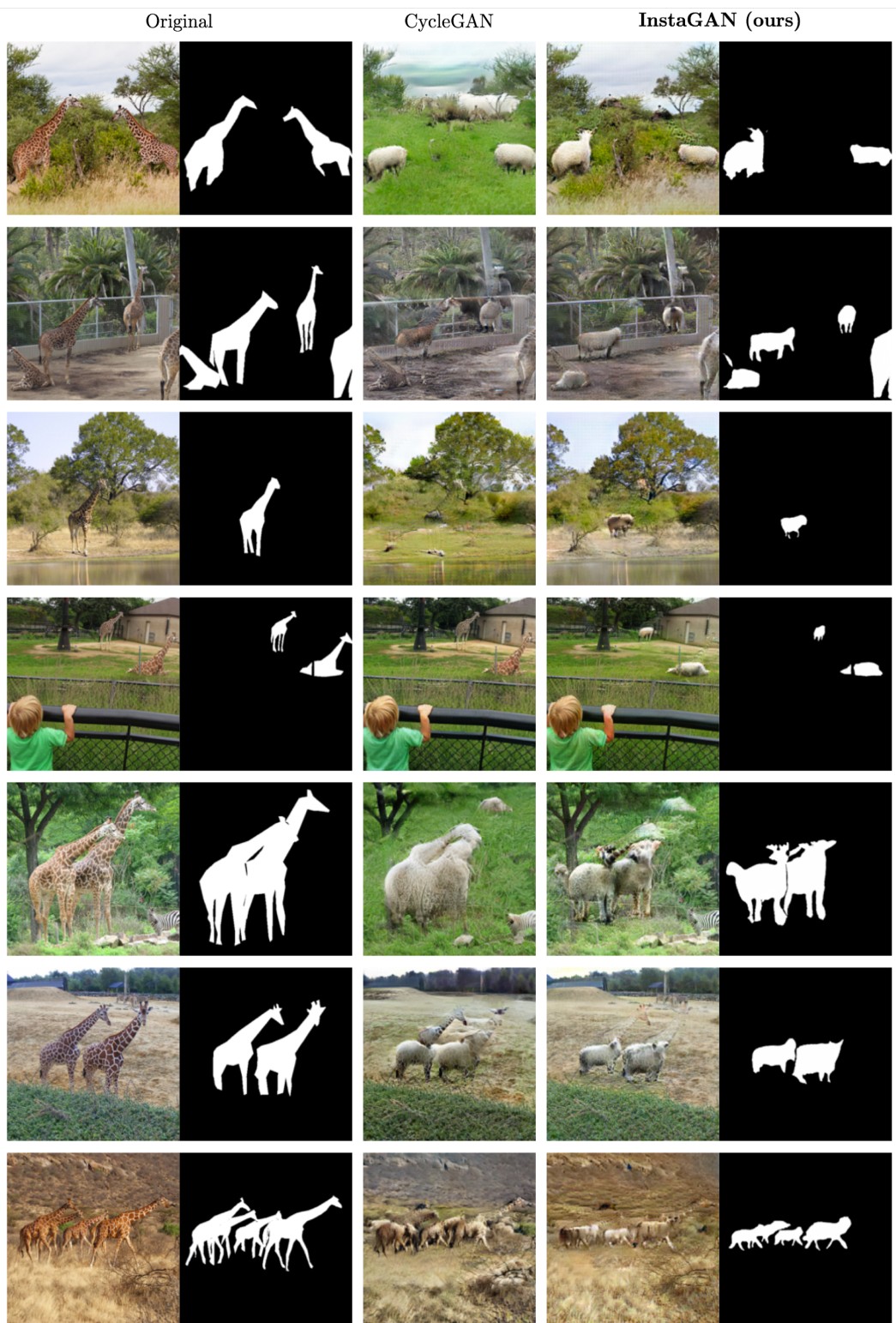

Figure 16: More translation results on COCO dataset (giraffe→sheep).

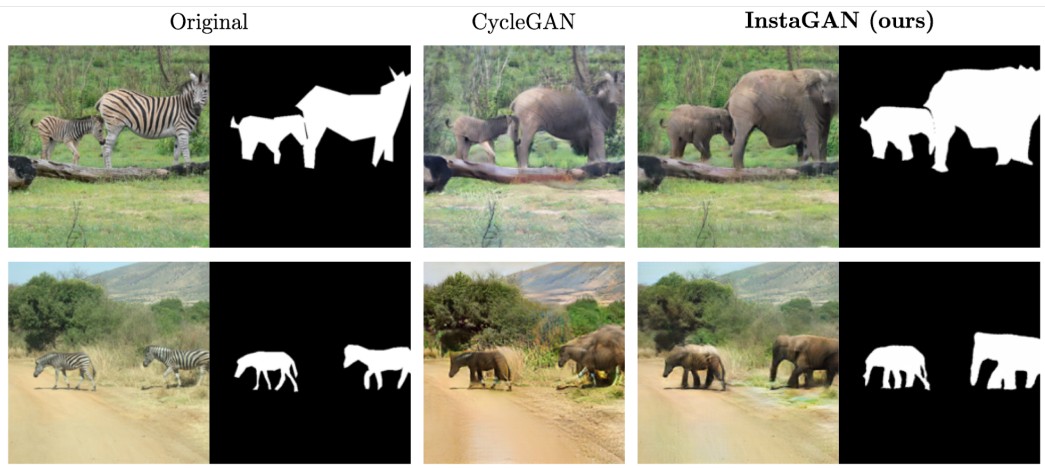

Figure 17: More translation results on COCO dataset (zebra→elephant).

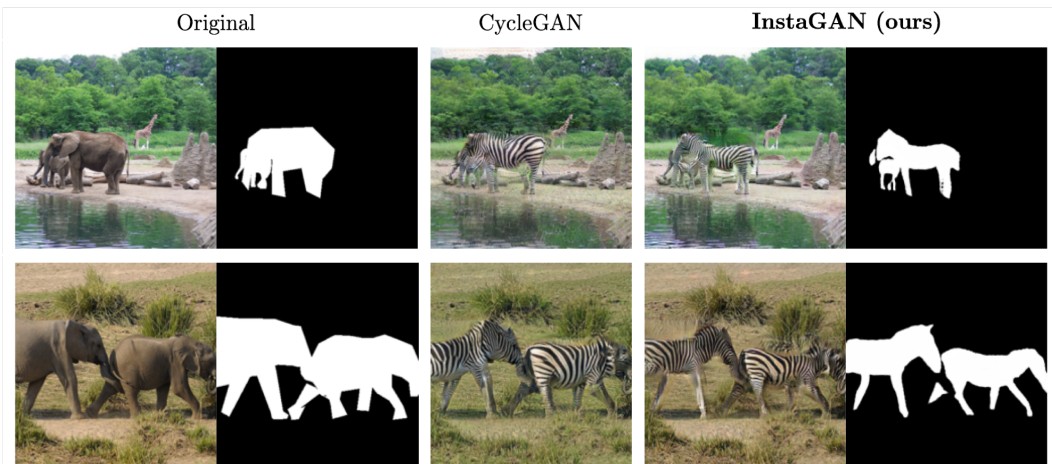

Figure 18: More translation results on COCO dataset (elephant→zebra).

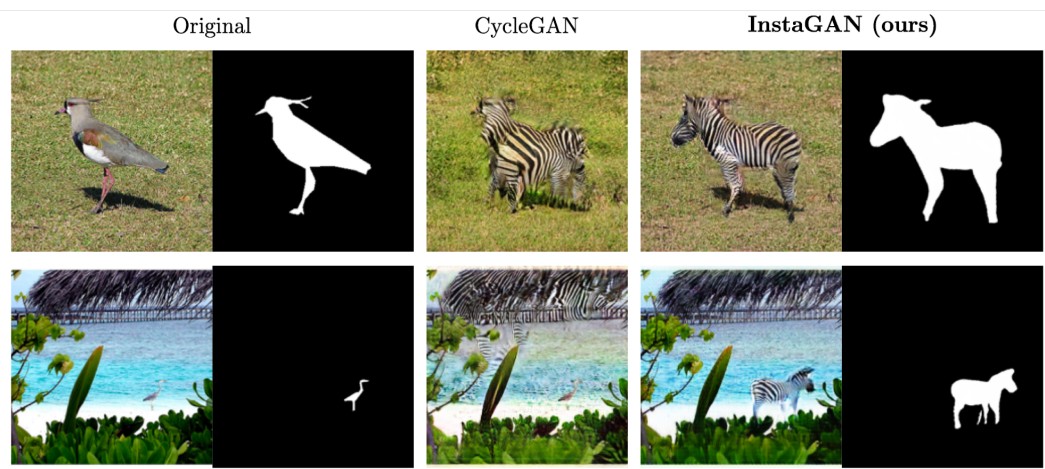

Figure 19: More translation results on COCO dataset (bird→zebra).

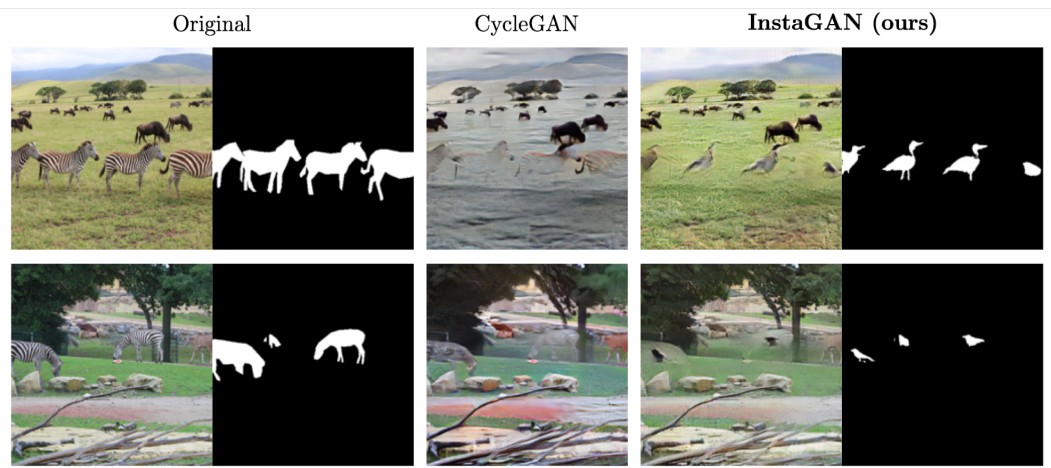

Figure 20: More translation results on COCO dataset (zebra→bird).

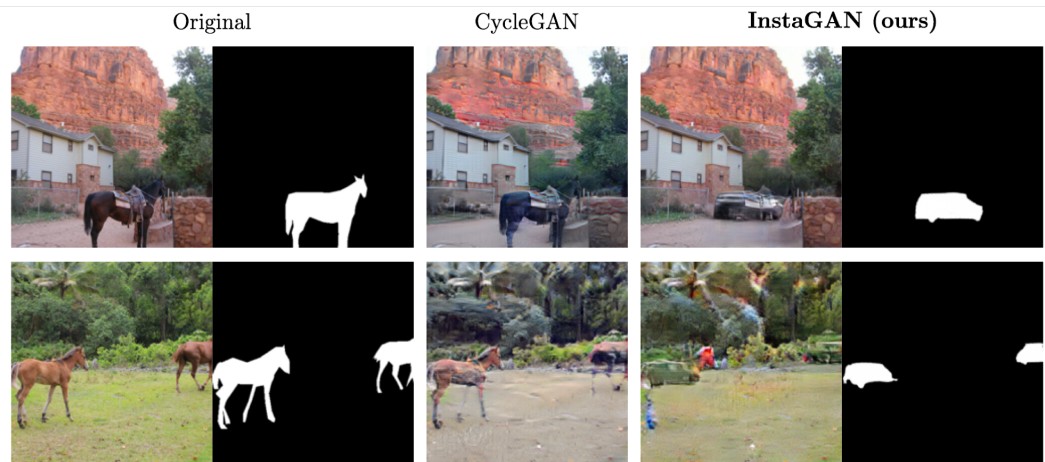

Figure 21: More translation results on COCO dataset (horse→car).

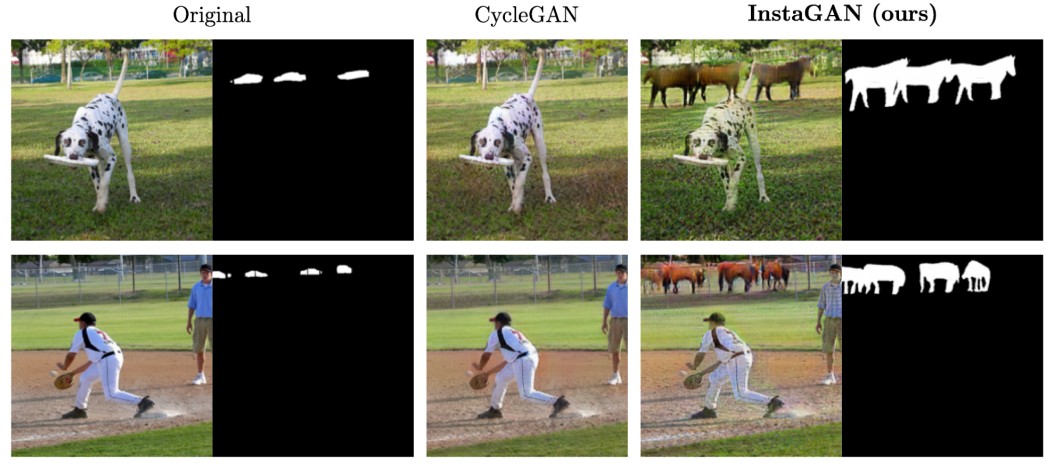

Figure 22: More translation results on COCO dataset (car→horse).

## F   MORE COMPARISONS WITH CYCLEGAN+SEG

To demonstrate the effectiveness of our method further, we provide more comparison results with CycleGAN+Seg. Since CycleGAN+Seg translates all instances at once, it often (a) fails to translate instances, or (b) merges multiple instances (see Figure 23 and 25), or (c) generates multiple instances from one instance (see Figure 24 and 26). On the other hand, our method does not have such issues due to its instance-aware nature. In addition, since the unioned mask losses the original shape information, our instance-aware method produces better shape results (*e.g.*, see row 1 of Figure 25).

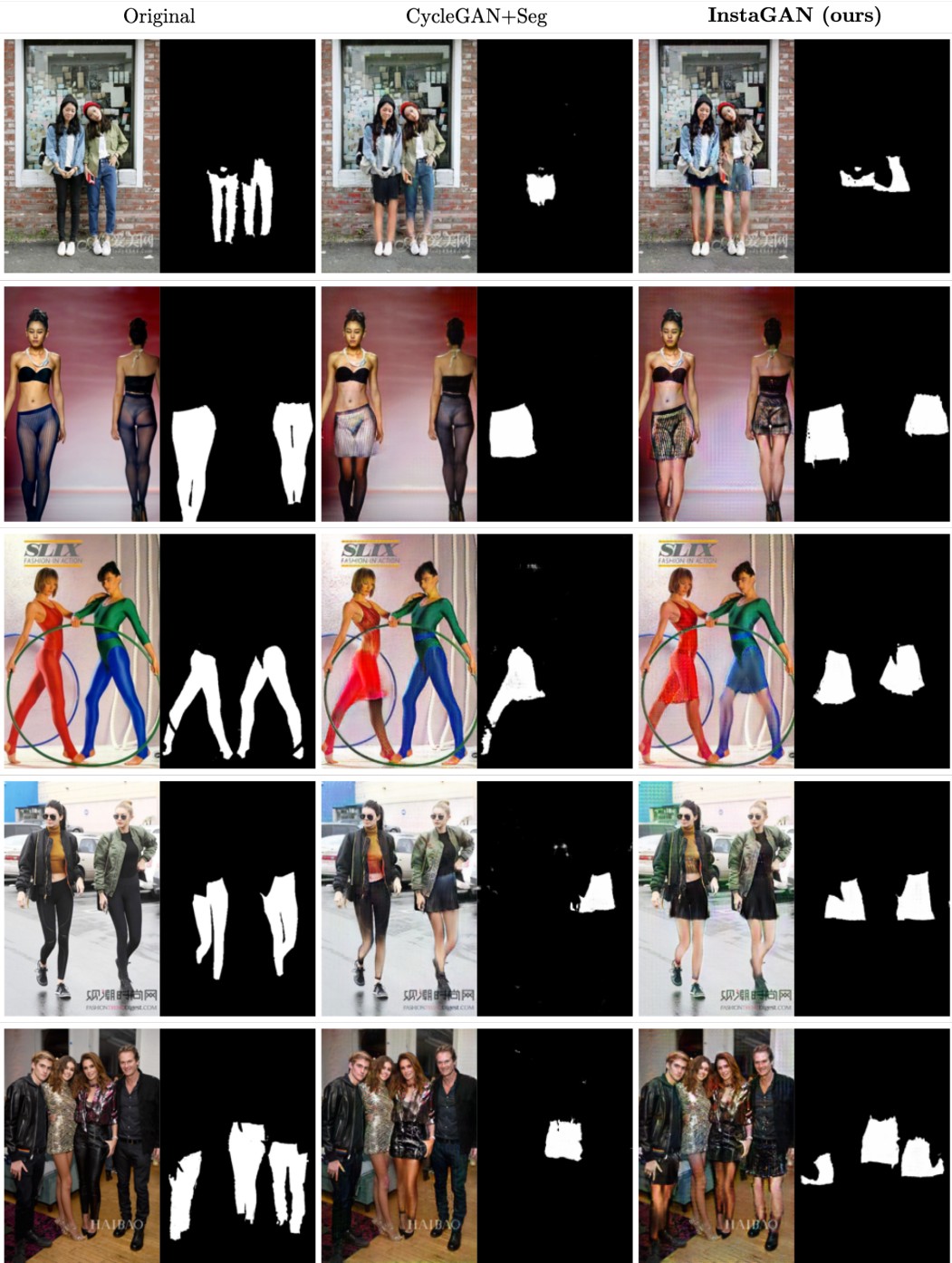

Figure 23: Comparisons with CycleGAN+Seg on MHP dataset (pants→skirt).

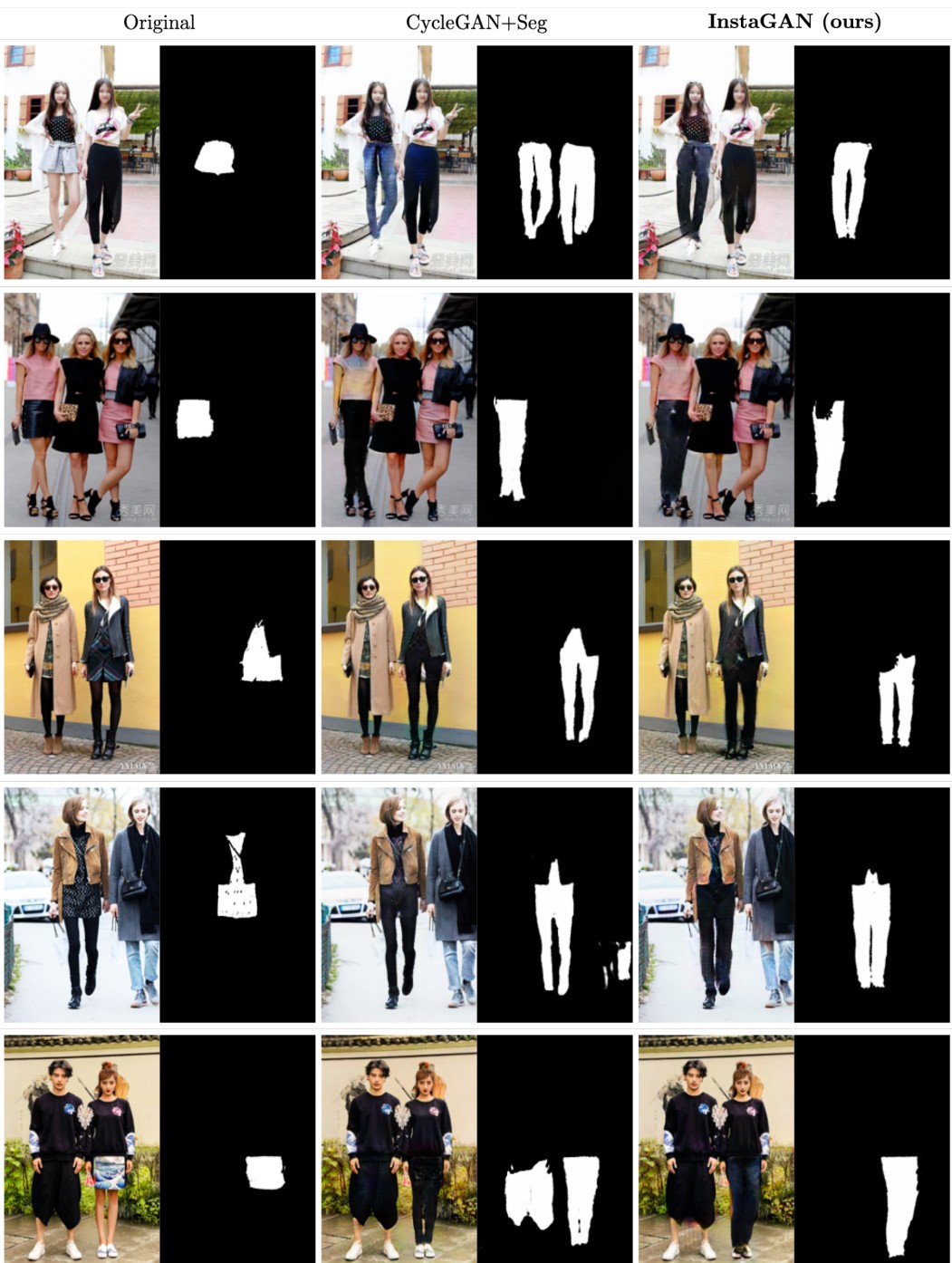

Figure 24: Comparisons with CycleGAN+Seg on MHP dataset (skirt→pants).

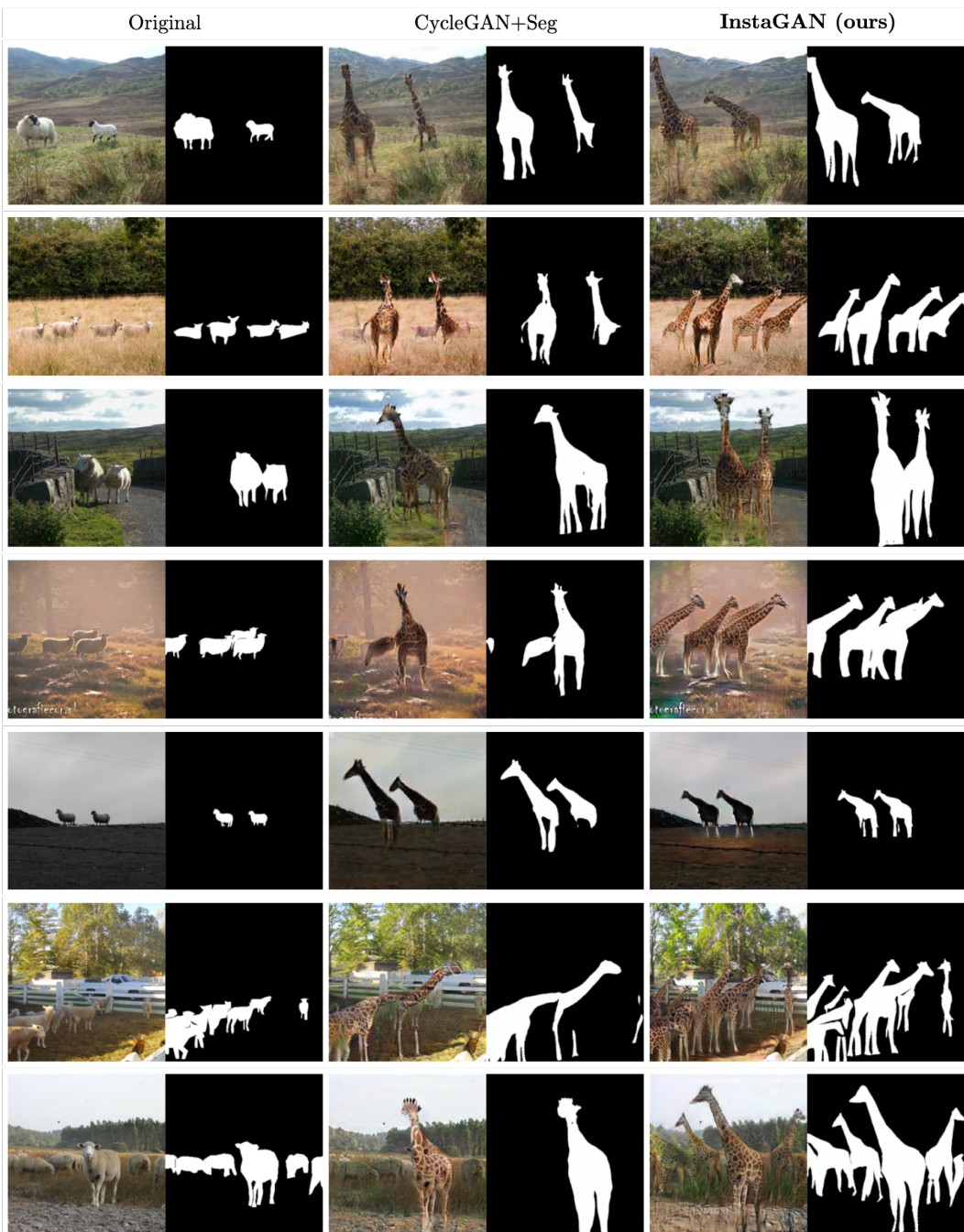

Figure 25: Comparisons with CycleGAN+Seg on COCO dataset (sheep→giraffe).

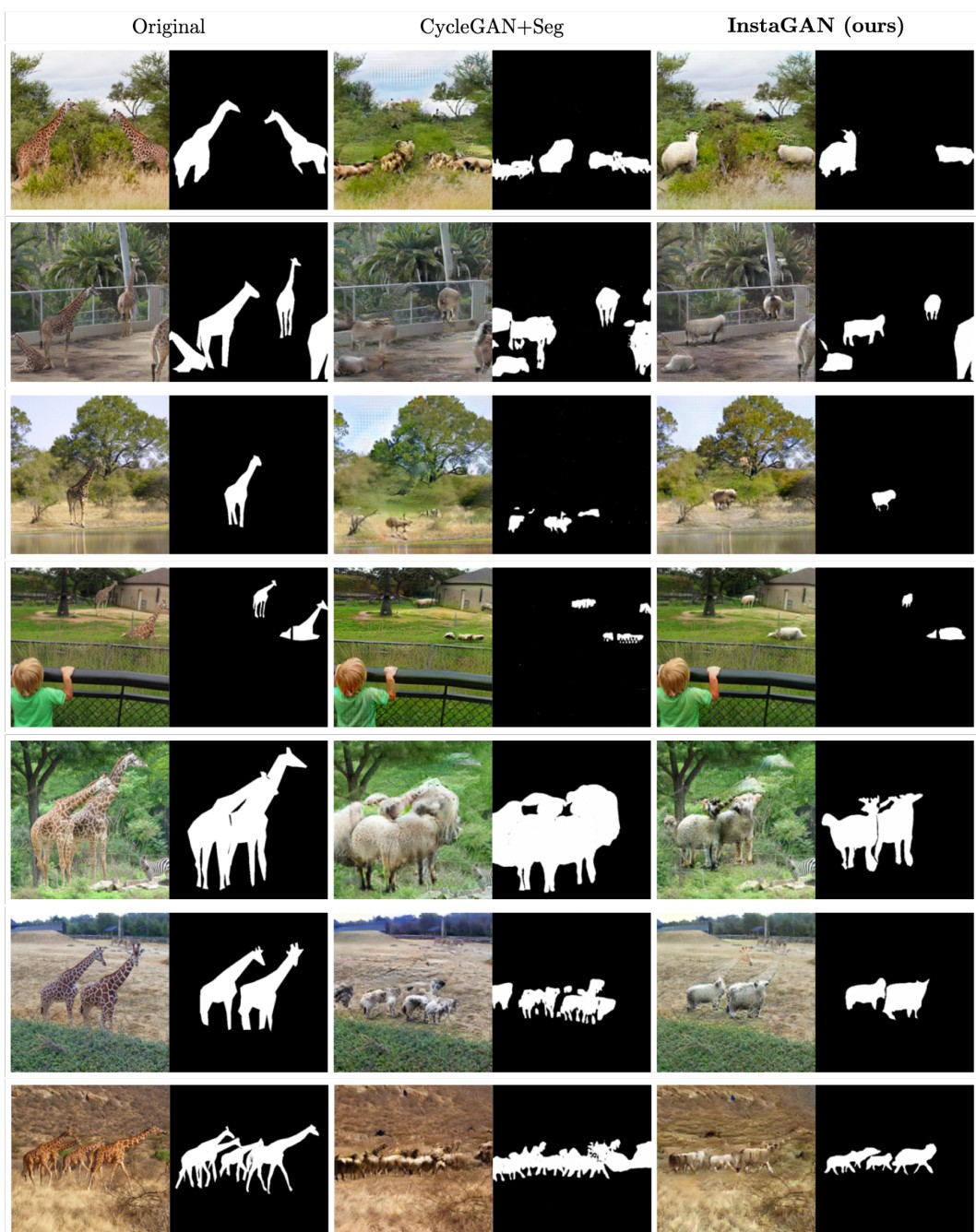

Figure 26: Comparisons with CycleGAN+Seg on COCO dataset (giraffe→sheep).

## G    GENERALIZATION OF TRANSLATED MASKS

To show that our model generalizes well, we searched the nearest training neighbors (in $L_2$-norm) of translated target masks. As reported in Figure 27, we observe that the translated masks (col 3,4) are often much different from the nearest neighbors (col 5,6). This confirms that our model does not simply memorize training instance masks, but learns a mapping that generalizes for target instances.

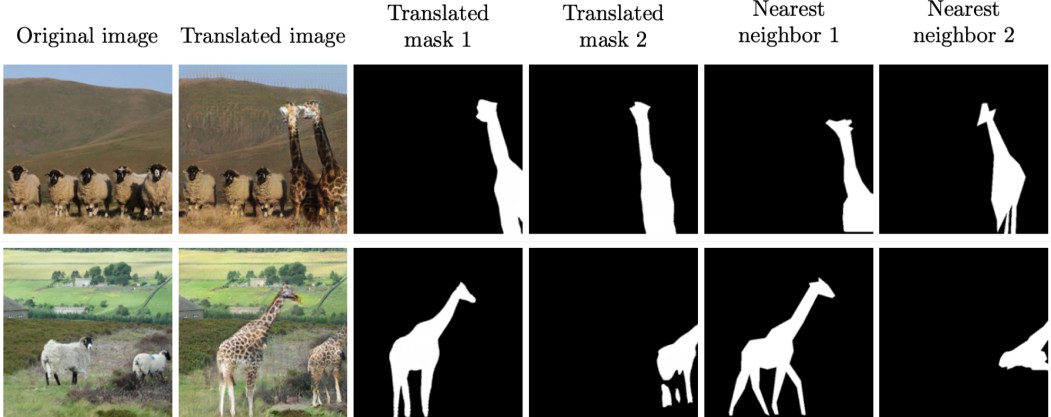

Figure 27: Nearest training neighbors of translated masks.

## H    TRANSLATION RESULTS OF CROP & ATTACH BASELINE

For interested readers, we also present the translation results of the simple crop & attach baseline in Figure 28, that find the nearest neighbors of the original masks from target masks, and crop & attach the corresponding image to the original image. Here, since the distance in pixel space (*e.g.*, $L_2$-norm) obviously does not capture semantics, the cropped instances do not fit with the original contexts as well.

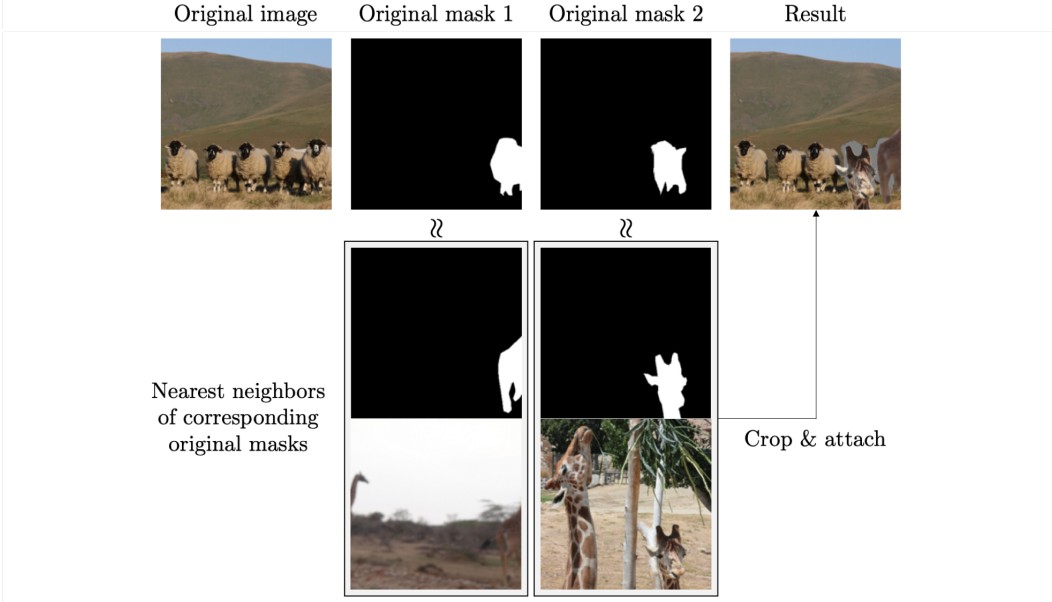

Figure 28: Translation results of crop & attach baseline.

# I VIDEO TRANSLATION RESULTS

For interested readers, we also present video translation results in Figure 29. Here, we use a predicted segmentation (generated by a pix2pix (Isola et al., 2017) model as in Figure 8 and Figure 12) for each frame. Similar to CycleGAN, our method shows temporally coherent results, even though we did not used any explicit regularization. One might design a more advanced version of our model utilizing temporal patterns *e.g.*, using the idea of Recycle-GAN (Bansal et al., 2018) for video-to-video translation, which we think is an interesting future direction to explore.

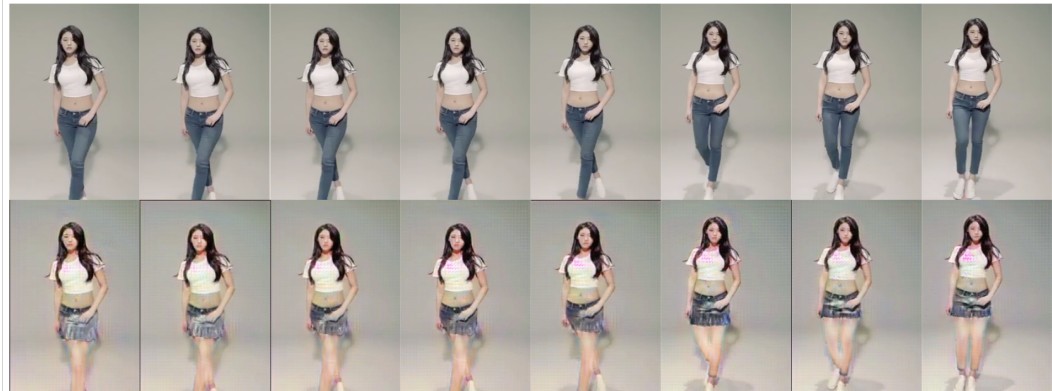

Figure 29: Original images (row 1) and translated results of our method (row 2) on a video searched from YouTube. We present translation results on successive eight frames for visualization.

## J    RECONSTRUCTION RESULTS

For interested readers, we also report the translation and reconstruction results of our method in Figure 30. One can observe that our method shows good reconstruction results while showing good translation results. This implies that our translated results preserve the original context well.

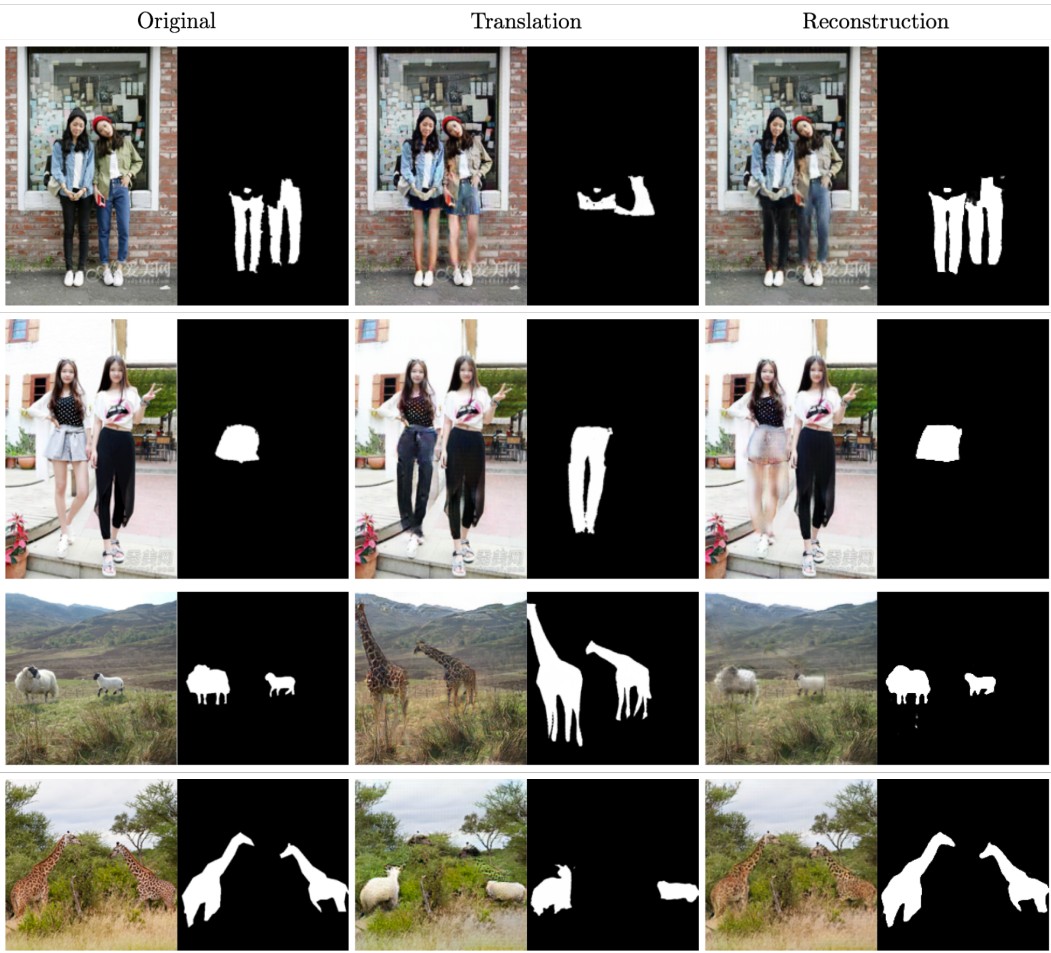

Figure 30: Translation and reconstruction results of our method.

