# OpenReview forum: "InstaGAN: Instance-aware Image-to-Image Translation"
_ICLR.cc/2019/Conference_

### Official Review · AnonReviewer2 · 2018-10-25
**Interesting idea, comparisons need to be improved**

**Rating:** 7
**Confidence:** 5

**Review:**

This paper proposes a well-designed instance level unsupervised image-to-image translation method which can handle the arbitrary number of instances in a permutation-invariant way. The idea is interesting and the results on various translation datasets are reasonable.

Pros:
* The proposed method process each instance separately to handle multiple instances. The summarization operation is a simple but effective way to achieve the permutation-invariant property. The context preserving loss is suitable for preserving the background information.
* The paper is well written and easy to follow.

Cons:
* My main concern is about the comparisons with CycleGAN in Figure 4 to 6. Although the CycleGAN+Seg results are shown in Figure 9 indicating that the proposed method can handle multiple instances better. I think there should also be CycleGAN+Seg results in Figure 4 to 6, since the instance segmentation is an extra information. And in my opinion, the CycleGAN+Seg can handle the situation where there are only a few instances (also can be observed in the 1st row in Figure 9). Besides, CycleGAN+Seg can naturally handle the arbitrary number of instances without extra computation cost.

Questions:
*  I wonder what will happen if the network does not permutation-invariant. Except that the results will vary for different the input order, will the generated quality decrease? Since the order information may be useful for some applications.

Overall, I think the proposed method is interesting but the comparison should be fairer in Figure 4 to 6.

---

> ### Author Response · Authors · 2018-11-19
> **Response to AnonReviewer2**
>
> Thank you very much for your valuable comments. In what follows, we provide our response to them. In the revised draft, we mark our major revisions by “red”.
>
> 1. More comparisons with CycleGAN+Seg
>
> As you suggested, we added more results comparing our method and CycleGAN+Seg in Appendix F of the revised draft. As you mentioned, CycleGAN+Seg shows comparable results to ours if there exists only a single instance (see Figure 24). However, even for a few more (2 or 3) instances, our method shows better results. Since CycleGAN+Seg translates all instances at once, it often (a) fails to translate instances, (b) merges multiple instances to a single one (see Figure 23, 25), or (c) generates multiple instances from a single instance (see Figure 24, 26). In addition, since the unioned mask often loses the original shape information, our instance-aware method may produce better shape results (e.g., see row 1 of Figure 25).
>
> 2. Non-permutation-invariant setting
>
> We first emphasize that our main focus is handling a set of permutation-invariant instance attributes. Under the setting, our proposed architecture is quite natural and conceptually simple. For applications where instances have some order information, one can simply translate instances sequentially following the order under our sequential mini-batch framework. One can also directly provide the order information as instance attributes. For example, if instances are sorted by depth, one can utilize depth map in addition to the instance masks as inputs of our network. We think that applying our method to non-permutation-invariant settings would be an interesting research direction.

---

> > ### Comment · AnonReviewer2 · 2018-11-24
> > **My main concern has been addressed**
> >
> > Thanks for your response and the revision. My may concern has been addressed. I suggest the authors also add these comparisons with CycleGAN+Seg to the main paper. I would like to raise my rating from 6 to 7.

---

### Official Review · AnonReviewer1 · 2018-11-02
**Nice formulation, good results!**

**Rating:** 8
**Confidence:** 5

**Review:**

Post rebuttal: I am satisfied by the points mentioned by authors!

----------------------------------------------------------------
Summary: The paper proposes to add instance-aware segmentation masks for the problem of unpaired image-to-image translation. A new formulation is proposed to incorporate instance masks with an input image to generate a new target image and corresponding mask. The authors demonstrate it on multiple tasks, and show nice results for each of them.

Pros:

1. The formulation is intuitive and well done!

2. The idea of sequential mini-batch translation connects nicely to the old school of making images by layering.

3. Nice qualitative analysis, and good results in comparison with Cycle-GAN (an obvious baseline for the formulation). I would make an observation that two domains for translation (such as sheep to giraffe, jeans to skirts etc) are thoughtfully selected because Cycle-GAN is somewhat bound to fail on them. There is no way Cycle-GAN can work for jeans to skirts because by design the distribution for images from both set would be mostly similar, and it is way too hard for the discriminator to distinguish between two. This ultimately leads the generator to act as an identity mapping (easily observed in all the qualitative examples).

4. The proposed approach can easily find direct application in places where a user-control is required for image editing or synthesis.

5. The literature review is extensive.

Cons:

1. My biggest criticism of this work is the absence of simple baselines.  Given the fact that the formulation use an instance segmentation map with the given input, the following obvious baseline need consideration:

Suppose the two domains are sheep and giraffe:

a. given the input of sheep and its instance mask, find a shape/mask in giraffe from the training images that is closest (it could be same location in image or some other similarity measure).

b. mask the input image using the sheep mask. Use giraffe mask and add corresponding RGB components of the masked giraffe (from the training set) to the masked input image.

The above step would give a rough image with some holes.

c. To remove holes, one can either use an image inpainting pipeline, or can also simply use a CNN with GAN loss.

I believe that above pipeline should give competitive (if not better) outputs to the proposed formulation. (Note: the above pipeline could be considered a simpler version of PhotoClipArt from Lalonde et al, 2007).

2. Nearest neighbors on generated instance map needs to be done. This enables to understand if the generated shapes are similar to ones in training set, or there are new shapes/masks being generated. Looking at the current results, I believe that generated masks are very similar to the training instances for that category. And that makes baseline described in (1) even more important.

3. An interesting thing about Cycle-GAN is its ability to give somewhat temporally consistent (if not a lot) -- ex. Horse to Zebra output shown by the authors of Cycle-GAN. I am not sure if the proposed formulation will be able to give temporally consistent output on shorts/skirts to jeans example. It would be important to see how the generated output looks for a given video input containing a person and its segmentation map  of jeans to generate a video of same person in shorts?

---

> ### Author Response · Authors · 2018-11-19
> **Response to AnonReviewer1**
>
> Thank you very much for your valuable comments. In what follows, we provide our response to them. In the revised draft, we mark our major revisions by “red”.
>
> 1. Simple crop & attach baseline
>
> As you suggested, in the revised draft, we report the translation results of the crop & attach baseline in Appendix H. The main problem of simply cropping and attaching the closest target instance would be that the translated results may lose the original contexts. For example, Figure 6 shows that the translated sheep/giraffes by our method have consistent poses (left, right, front), while the simple baseline cannot do it. Here, since the distance in pixel space (e.g., L2-norm) obviously does not capture semantics, the cropped instances do not fit with the original contexts either. We finally remark that our framework can also utilize other instance attributes, e.g., color or depth, which often cannot be simply cropped & attached.
>
> 2. Memorization issue
>
> To address your concern on whether generated masks are very similar to the training instances, we searched nearest training neighbors (in L2-norm) of translated target masks and report them in Appendix G of the revised draft. We observe that the translated masks are often much different from the nearest neighbors. The results also support that the simple crop & attach baseline would not work for our problem. More fundamentally, as it has been well evidenced in the literature that GAN generalizes well, we also strongly believe that our GAN-based method also does, i.e., generated masks are not just copies (or something close to them) of the training instances.
>
> 3. Temporal coherency
>
> We think applying our method to video-to-video translation is definitely interesting as you suggested. We report the video translation results in Appendix I of the revised draft. Here, we use a predicted segmentation (generated by a pix2pix model as in Figure 7 and Figure 12) for each frame. Similar to CycleGAN, our method shows temporally coherent results, even though we did not use any explicit temporal regularization.
>
> Here, we remark that one can even enforce temporal coherence to our model explicitly, which is an interesting future research direction. For example, one can consider video segmentation networks [1] (instead of pix2pix) to predict temporally coherent instance segmentations. One might design a more advanced version of our model utilizing temporal patterns, e.g., using the idea of Recycle-GAN [2] for video-to-video translation.
>
> [1] DAVIS: Densely Annotated VIdeo Segmentation Challenge (https://davischallenge.org)
> [2] Bansal et al. Recycle-GAN: Unsupervised Video Retargeting. ECCV 2018.

---

### Official Review · AnonReviewer3 · 2018-11-11
**well-written paper, nice method, somewhat limited results+evaluation**

**Rating:** 7
**Confidence:** 4

**Review:**

This paper does unpaired cross-domain translation of multi-instance images, proposing a method -- InstaGAN -- which builds on CycleGAN by taking into account instance information in the form of per-instance segmentation masks.

=====================================

Pros:

The paper is well-written and easy to understand. The proposed method is novel, and does a good job of handling a type of information that previous methods couldn’t.

The motivation for each piece of the model and training objective is clearly explained in the context of the problem. Intuitively seems like a nice and elegant way to take advantage of the extra segmentation information available.

The results look pretty good and clearly compare favorably with CycleGAN and other baselines. The tested baselines seem like a fair comparison -- for example, the model capacity of the baseline is increased to compensate for the larger proposed model.

=====================================

Cons / suggestions:

The results are somewhat limited in terms of the number of domains tested -- three pairs of categories (giraffe/sheep, pants/skirts, cup/bottle).  In a sense, this is somewhat understandable -- one wouldn’t necessarily expect the method to be able to translate between objects with different scale or that are never seen in the same contexts (e.g. cups and giraffes). However, it would still have been nice to see e.g. more pairs of animal classes to confirm that the category pairs aren’t the only ones where the method worked.

Relatedly, it would have been interesting to see if a single model could be trained on multiple category pairs and benefit from information sharing between them.

The evaluation is primarily qualitative, with quantitative results limited to Appendix D showing a classification score. I think there could have been a few more interesting quantitative results, such as segmentation accuracy of the proposed images for the proposed masks, or reconstruction error. Visualizing some reconstruction pairs (i.e., x vs. Gyx(Gxy(x))) would have been interesting as well.

I would have liked to see a more thorough ablation of parts of the model. For example, the L_idt piece of the loss enforcing that an image in the target domain (Y) remain identical after passing through the generator mapping X->Y. This loss term could have been included in the original CycleGAN as well (i.e. there is nothing about it that’s specific to having instance information) but it was not -- is it necessary?

=====================================

Overall, while the evaluation could have been more thorough and quantitative, this is a well-written paper that proposes an interesting, well-motivated, and novel method with good results.


==========================================================================

REVISION

The authors' additional results and responses have addressed most of my concerns, and I've raised my rating from 6 to 7.

> We remark that the identity mapping loss L_idt is already used by the authors of the original CycleGAN (see Figure 9 of [2]).

Thanks, you're right, I didn't know this was part of the original CycleGAN. As a final suggestion, it would be good to mention in your method section that this loss component is used in the original CycleGAN for less knowledgeable readers (like me) as it's somewhat hard to find in the original paper (only used in some of their experiments and not mentioned as part of the "main objective").

---

> ### Author Response · Authors · 2018-11-19
> **Response to AnonReviewer3**
>
> Thank you very much for your valuable comments. In what follows, we provide our response to them. In the revised draft, we mark our major revisions by “red”.
>
> 1. More translation results
>
> Following your suggestion, we report additional translation results in Appendix E of the revised draft. In particular, we additionally report the results for zebra<->elephant, bird<->zebra, and car<->horse. For the case of zebra<->elephant (which is the easiest translation task among three), both CycleGAN and our method succeed to translate images, but ours shows better details (see Figure 17 and Figure 18). For other cases of bird<->zebra and car<->horse, our method succeeds to translate, while CycleGAN fails, i.e., generates target textures in random location (see Figure 19), remove instances (see Figure 20) or learns an identity mapping (see Figure 21 and 22).
>
> 2. Extension to multiple domains
>
> We remark that our model is directly extendable to many-to-many domain transfer settings using the idea of StarGAN [1]. We definitely believe that exploring this direction would be an interesting future research direction.
>
> [1] Choi et al. StarGAN: Unified Generative Adversarial Networks for Multi-Domain Image-to-Image Translation. CVPR 2018.
>
> 3. Quantitative results and visualizing some reconstruction pairs
>
> You suggested to measure the segmentation accuracy and the reconstruction error as more quantitative results. First, in our unpaired setting, the segmentation accuracy of the translated images is hard to be measured since we do not have ground-truth segmentation labels for generated images. Next, we think that the reconstruction error is not an ideal evaluation metric as it only measures the loss in the original context, but does not capture the translation performance. Instead, in Appendix J (see Figure 30) of the revised draft, we report new qualitative results showing that our method has both good reconstruction and translation results (while CycleGAN fails to translate for the same images as in Figure 13-16). The results confirms that our translated results indeed preserve the original context well.
>
> 4. Identity mapping loss
>
> We remark that the identity mapping loss L_idt is already used by the authors of the original CycleGAN (see Figure 9 of [2]). Hence, in our implementation of CycleGAN, we used it as well. We emphasize that our contribution is not on the identity mapping loss, but the three new components: permutation-invariant architecture, context preserving loss, and sequential translation technique. We indeed report ablation study for all the components in Figure 9 (and more detailed ablation for sequential translation in Figure 10).
>
> [2] Zhu et al. Unpaired Image-to-Image Translation using Cycle-Consistent Adversarial Networks. ICCV 2017.

---

> > ### Comment · AnonReviewer3 · 2018-12-07
> > **Thanks for the responses**
> >
> > Thanks for your responses. I've updated my review above.

---

### Public Comment · ~zhiyi_cao1 · 2018-10-08
**Novel work！ With a limited GPU memory question。**

With a limited GPU memory and en- hances the network to generalize better for multiple instances? Why?

Can you open your code?

---

> ### Author Response · Authors · 2018-10-09
> **Thanks for your interest**
>
> Due to our goal for handling many instances, the number of backpropagation paths (and the required GPU memory during training) linearly increases with respect to the number of input instances. Hence, given the memory, one can train a limited number of instances, and thus the learned model might suffer from poor generalization for test samples containing a large number of instances (see the first paragraph in Section 2.3). The issue is expected to be more severe for higher resolution images, as they require more backpropagation memory.
>
> To address the issue, we proposed a sequential mini-batch technique (see Section 2.3), which allows to train samples of arbitrary many instances without increasing the memory. Consequently, it improves the testing performance for many instances (see 2nd, 3rd row of Figure 9 and 10). Furthermore, it even improves the performance of a few instances, due to its data augmentation effect (see 1st row of Figure 9 and 10).
>
> Due to the double blind policy, we currently plan to release our code after the paper decision.

---

> > ### Public Comment · ~zhiyi_cao1 · 2018-10-10
> > **When you test your model, do you need given the mask?**
> >
> > I want to know how do you create the mask. The mask is not given in the dataset your proposed.
> >
> > The sequential mini-batch training with instance subsets (mini-batches) of size 1,2                   ,it is difficult to understand!
> >
> > I think your algorithm is much more complicated than cyclegan, especially the sequential mini-batch training part.

---

> > > ### Author Response · Authors · 2018-10-11
> > > **Thank you very much for your additional comments**
> > >
> > > In this paper, we primarily assume that all images are annotated by the corresponding segmentation masks (we use such datasets including CCP, MHP and COCO in our experiments). All our contributions are toward how to utilize such additional information effectively for complicated image-to-image translation tasks.
> > >
> > > Nevertheless, for reducing the annotation cost, one may suggest to use predicted masks instead of real ones (see the second paragraph of Section 3.1). This is not our main focus, but we also show that our approach has potential to work well under the artificial masks (see Figure 8 and Figure 12 in Appendix). We think exploring how to train our models without such additional annotations is an interesting research direction in the future.
> > >
> > > “The sequential mini-batch training with instance subsets (mini-batches) of size 1, 2 and 1” in the caption of Figure 3 means that the input image contains total 4 instances, and we divided them into three subsets a_1, a_2, a_3 of size |a_1|=1, |a_2|=2, and |a_3|=1, respectively. Namely, we use the subset a_i for the i-th iteration, and at the final iteration (i=3), our model produces a result that all 4 instances are translated.
> > >
> > > We also remark that all the newly proposed ideas (including the sequential mini-batch training) are conceptually intuitive and simple. Hence, the implementation is straightforward upon the original CycleGAN code. As we mentioned earlier, we plan to release our code after the paper decision.

---

### Author Response · Authors · 2018-11-20
**Summary of First Revision**

Dear Reviewers,

In the first revision, we provide additional experimental results requested by all reviewers in Appendix E, F, G, H, I and J, which are highlighted by "red" texts. We think that the revised draft can now incorporate a broader range of interests of readers, and also would provide useful information to guide a future interesting research direction.

We strongly believe that we made an important and novel step toward solving the complex cross-domain generation problem.

If you have any further questions or suggestions, please do not hesitate to let us know.

Many thanks again for all your sincere contributions on ICLR 2019,
Authors

---

### Meta-Review · Area_Chair1 · 2018-12-15

**Confidence:** 5
**Recommendation:** Accept (Poster)

**Metareview:**

This paper addresses a promising method for unpaired cross-domain image-to-image translation that can accommodate multi-instance images. It extends the previously proposed CycleGAN model by taking into account per-instance segmentation masks. All three reviewers and AC agree that performing such transformation in general is a hard problem when significant changes in shape or appearance of the object have to be made, and that the proposed approach is sound and shows promising results. As rightly acknowledged by R1 ‘The formulation is intuitive and well done!’

There are several potential weaknesses and suggestions to further strengthen this work:
(1) R1 and R2 raised important concerns about the absence of baselines such as crop & attach simple baseline and CycleGAN+Seg. Pleased to report that the authors showed and discussed in their response some preliminary qualitative results regarding these baselines. In considering the author response and reviewer comments, the AC decided that the paper could be accepted given the comparison in the revised version, but the authors are strongly urged to include more results and evaluations on crop & attach baseline in the final revision if possible.
(2) more quantitative results are needed for assessing the benefits of this approach (R3). The authors discussed in their response to R3 that more quantitative results such as the segmentation accuracy of the synthesized images are not possible since no ground-truth segmentation labels are available. This is true in general for unpaired image-to-image translation, however collecting annotations and performing such quantitative evaluation could have a substantial impact for assessing the significance of this work and can be seen as a recommendation for further improvement.
(3) the proposed model performs translation for a pair of domains; extending the work to multi-domain translation like StarGAN by Choi et al 2018 or GANimation by Pumarola 2018 would strengthen the significance of the work. The authors discussed in their response to R3 that this is indeed possible.